# Multi-omic and single-cell profiling of chromothriptic medulloblastoma reveals genomic and transcriptomic consequences of genome instability

Petr Smirnov[1,2,15], Moritz J. Przybilla [2,3,4,15], Milena Simovic-Lorenz[1,15], R. Gonzalo Parra [2,3,5,15], Hana Susak[2,3,15], Manasi Ratnaparkhe[1], John KL. Wong[6], Verena Körber [7], Jan-Philipp Mallm [8], George Philippos[1,9], Martin Sill[10,11], Thorsten Kolb[1], Rithu Kumar[1], Nicola Casiraghi[2,3], Konstantin Okonechnikov[10,11], David R. Ghasemi [10,11,12], Kendra Korinna Maaß[10,11,12], Kristian W. Pajtler [10,11,12], Anna Jauch[13], Andrey Korshunov[14], Thomas Höfer [7], Marc Zapatka [6], Stefan M. Pfister [10,11,12], Wolfgang Huber [2], Oliver Stegle [2,2,3,16] ✉ & Aurélie Ernst[1,16] ✉

Chromothripsis is a frequent form of genome instability, whereby a presumably single catastrophic event generates extensive genomic rearrangements of one or multiple chromosome(s). However, little is known about the heterogeneity of chromothripsis across different clones from the same tumour, as well as changes in response to treatment. Here we analyse single-cell genomic and transcriptomic alterations linked with chromothripsis in human p53-deficient medulloblastoma and neural stem cells ($n = 9$). We reconstruct the order of somatic events, identify early alterations likely linked to chromothripsis and depict the contribution of chromothripsis to malignancy. We characterise subclonal variation of chromothripsis and its effects on extrachromosomal circular DNA, cancer drivers and putatively druggable targets. Furthermore, we highlight the causative role and the fitness consequences of specific rearrangements in neural progenitors.

Chromothripsis (CT) is a type of genome instability, by which a presumably single catastrophic event leads to substantial genomic rearrangements of one or a few chromosome(s)[1,2]. Generally considered as an early event in the evolution of a tumour, CT likely plays a causative role in the development of a number of tumours by generating multiple genomic aberrations simultaneously. In line with this, rearrangements due to CT were detected in more than 25% of cancer patients in two large pan-cancer studies[3,4]. In specific tumour types or molecular subgroups, the prevalence for CT reaches 100%, such as in medulloblastoma with germline *TP53* mutations (Li-Fraumeni syndrome, LFS), which is the focus of this study. As in a number of other

tumour types[2,5–7], CT is linked with poor prognosis for these patients, as compared to medulloblastomas from the same molecular subgroup without CT. In the context of *TP53* mutations in the germline, it is conceivable that multiple CT events may occur in different cells and most of them are not selected for and therefore undetected.

Longitudinal studies on the evolution of CT chromosomes between matched primary and relapsed tumours showed that CT patterns may be either (i) stabilised (ii) eliminated or (iii) undetected at initial diagnosis but present in the relapsed tumour[4,8,9]. Importantly, elimination as well as newly detected CT chromosomes suggest that a subset of tumour cells in the initial tumour may potentially lack or

already carry the CT chromosome. These findings question the paradigm that CT is a single early event in tumour development, which would imply that the CT chromosome would be present in the vast majority of the tumour cells. However, conclusive evidence of the extent to which CT varies across tumour cells and clones is missing. As CT was shown to drive tumour development through the activation of oncogenes and the disruption of tumour suppressor genes[4,10], clonal heterogeneity in CT could also have implications for the role of cancer drivers and therapeutic targets within CT tumours. In addition, CT was linked to compromised function of essential factors such as p53, ATM and critical DNA repair proteins, suggesting that the inactivation of specific pathways and checkpoints may facilitate the occurrence of CT events and/or the survival of the cells after such an event[2,11,12]. However, direct evidence of potential enabling mechanisms is limited.

Sequencing of cultured cells showed that processes such as mitotic errors, micronuclei formation, centromere inactivation, chromatin bridges but also telomere dysfunction can cause a range of rearrangements, including CT[13–17]. Although modelling CT in cell culture systems has allowed putative mechanisms to be proposed, the way in which spontaneous CT events occur in human cells remains largely unknown. It is unclear to which extent mechanisms derived from artificially inducing CT in vitro reflect CT events in human cancer. Single-cell DNA sequencing (scDNA-seq) studies in the context of CT are only beginning to emerge. Pellman and colleagues reported mechanistic insights into the generation of complex rearrangements from sequencing cultured clones and single cells from the retinal pigment epithelial (RPE1) cell line[16,17]. Korbel and colleagues investigated structural variation in the RPE1 cell line and showed complex rearrangements for one leukaemia sample as a proof-of-principle[18,19]. To study this process in situ, we set off to characterise the heterogeneity in CT patterns across tumours. Even though subclonality of CT was suggested by previous studies[20], CT has not been analysed in primary patient material and cells obtained from PDX models at single-cell resolution, and previous single-cell studies on medulloblastoma have not focused on CT[21–25].

Here, we leverage bulk and single-cell sequencing assays, combined with fluorescence in situ hybridisation (FISH), immunofluorescence analyses and CRISPR/Cas9 knockouts to investigate the origins and functional consequences of CT in LFS medulloblastoma. We generated shallow single-cell DNA- and single-cell RNA-seq profiles from 663 and 22,500 cells from 7 LFS medulloblastoma samples, respectively, including three brain tumours and four PDX samples. We demonstrate the ability to detect CT events using single-cell DNA-seq in tumours, further unravelling the extent of intra-tumour heterogeneity with clonal resolution. In addition, we highlight potential mechanisms for the formation of extrachromosomal circular DNA (ecDNA). Using matched single-cell RNA-seq (scRNA-seq) data, we characterise the malignant cell types and investigate differences to non-CT medulloblastoma. By integrating scDNA-seq and scRNA-seq information based on somatic copy number profiles that can be identified from both data modalities, we shed light into potential transcriptomic consequences of CT and its impact on tumour evolution. Finally, we identify a putative role for the SETD2 methyltransferase in the early stages of the development of CT medulloblastomas using functional analyses in neural stem cells.

## Results

### Rearrangements due to CT can be detected in single tumour clones

We explored how CT contributes to inter cell genetic heterogeneity and generates oncogenic drivers that increase cell fitness and tumour aggressiveness. We performed single-cell DNA and RNA-seq (single-cell and single-nuclei sequencing; hereafter termed single-cell (sc) seq) of paediatric medulloblastomas with CT that carry a germline *TP53* variant (LFS, n = 7, including 3 patient tumours and 4 patient-derived

xenograft (PDX) models, primary and relapsed, see Fig. 1 & Supplementary Data 1). Medulloblastomas in LFS patients constitute a paradigm for the understanding of this phenomenon, as CT is present in close to 100% of these cancers[2,26]. This patient collective is embedded in a larger population-scale deep sequencing cohort of patients that span all molecular subgroups of medulloblastoma (n = 227).

### Subclonal CT events contribute substantially to intra-tumour heterogeneity in LFS medulloblastoma

With CT being present in all LFS medulloblastomas analysed so far[2,26], we asked whether we could characterise the underlying subclonal heterogeneity using single-cell sequencing. Briefly, to identify subclones and their copy number variation (CNV) profiles, we first estimated total copy number in larger genomic segments in each cell (500 kb to 1 Mb; using scAbsolute[27], followed by clustering). This identified between 1 and 5 distinct genetic clones per sample (Fig. 2a, b, Supplementary Fig. 1–7). Next, to study clonal heterogeneity in CT, we called CNV profiles using aggregate read counts for each clone, which provided higher resolution (20 kb; Fig. 2a) and enabled the identification of CT regions (based on density signature of copy number state switches[3,4]; "Methods"). We assessed the consistency of the resulting CT estimates across clones with matched bulk WGS data[3], finding that 85% of CT regions identified by either approach replicated in the other one (Fig. 2a, Supplementary Data 2). We also assessed the sensitivity of our strategy to detect CT events as a function of sequencing coverage (using downsampling; Supplementary Fig. 8a), and we applied our workflow to an independent single-cell dataset from a cell line with an induced CT event, confirming high specificity (Supplementary Fig. 8b). Finally, we assessed the consistency of patterns of oscillating CNV changes, as characteristics for CT, across single-cell DNA-seq technologies in PDX models, and found evidence for precursor cells in the corresponding primary tumour (Supplementary Fig. 9).

Having confirmed the accuracy of the CT identification based on scDNA-seq, we next set out to study broad patterns of subclonal CT. Across samples, we identified both clonal and subclonal CT regions (considering 4/6 samples with two or more clones, Fig. 2b, c). For example, in LFS-MBP Nuclei, we identified clonal CT regions on chromosomes 4, 7, 16 and X and subclonal events on chromosomes 5, 12, 14, 17, and 19 (Fig. 2a). While all but one of these CT events (chromosome 12) were also detected in bulk WGS data, bulk profiling cannot discriminate between clonal or subclonal CT events. On the level of individual CNV events, as expected, the majority of events overlapped one of the identified CT regions (62%), reflecting the high density of copy number breakpoints caused by CT. This was despite non-CT associated CNVs altering a considerably larger fraction of the genome (24% non-CT; 7% CT associated), highlighting the specific role of CT in driving a large number of CNV breakpoints (e.g., chromosome 7 inset Fig. 2a). Similar patterns were observed across the full dataset, with 36-67% of the CNV events overlapping CT regions (Fig. 2d). These findings are in line with data from Notta et al.[28] from bulk analyses in pancreatic cancer, supporting the notion that the majority of the overall genome instability in CT tumours can be attributed to a small number of CT events.

Critically, the role of CT as driver of CNV events was also evident when considering subclonal CNVs. Across samples, between 43% and 45% of all subclonal CNV events were overlapping with CT regions (Fig. 2e). Collectively, our data underline a broader relevance of CT, not only as a driver of tumour instability per se, but specifically also as a driver of a substantial fraction of subclonal genomic alterations, thereby extending previous knowledge derived from bulk data that could not differentiate between clonal and subclonal CT[28].

### Chromothripsis is a major event for the formation of extrachromosomal circular DNA structures (ecDNAs)

Extrachromosomal circular DNA (ecDNA) fragments carrying amplified oncogenes were previously suggested to be generated by CT[2,10]

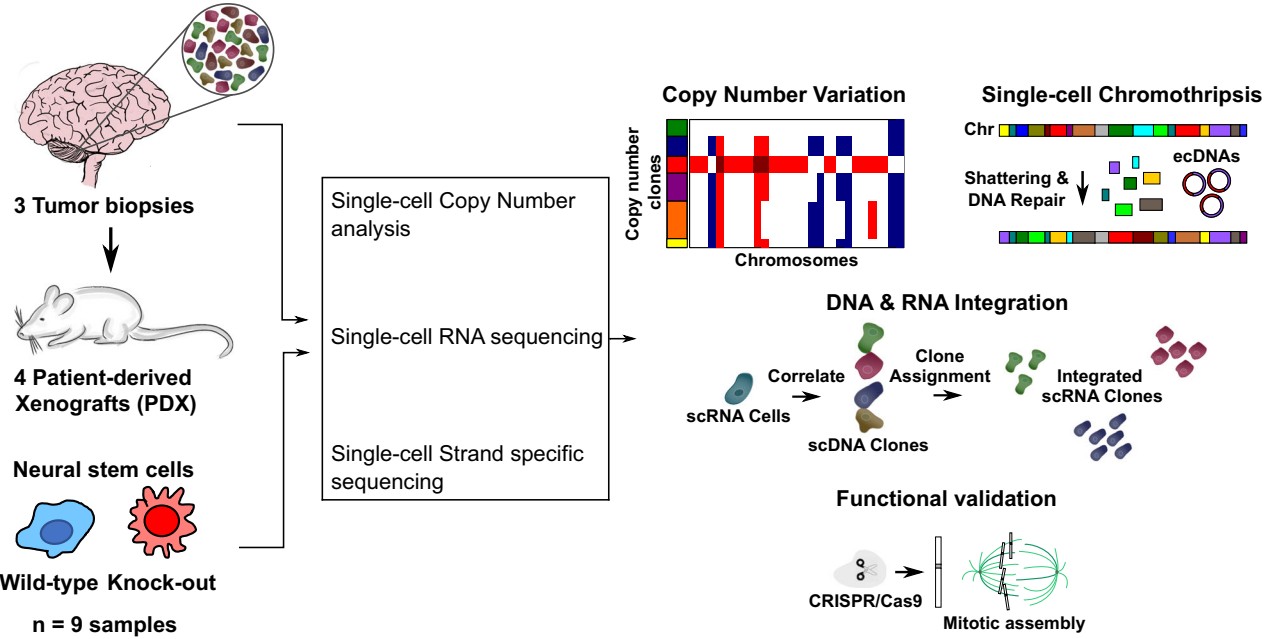

**Fig. 1 | Overview of the samples and workflow of this study.** Left, overview of the biological samples that were analyzed and of the methods that were applied. Right, overview of the data analysis workflow.

but have not been studied extensively at the single-cell level in tumours. Seminal studies, primarily in cell lines, have characterised the evolutionary dynamics of ecDNAs[29], segregation patterns[30], structural heterogeneity[31] and associated oncogene expression[32,33]. However, the relationship between intra-tumour heterogeneity in ecDNAs and in CT is still unclear. To detect and quantify putative ecDNAs, we searched for small fragments of a few Mb highly amplified carrying oncogenes in the scDNA-seq data (Fig. 3a, sample MB243-Nuclei). By assembly from the matched bulk data, we confirmed that these amplified fragments were indeed circular DNA structures carrying oncogenes (e.g., *GLI2*, Fig. 3b). Interestingly, their copy number varied from 5 to more than 100 copies per nucleus in the single-cell data, suggesting an additional level of heterogeneity (Fig. 3c; "Methods").

We observed no correlation between the number of copy-number segments per chromosome detected in each single cell and the copy number of ecDNAs, with some cells with only 4 or 5 breakpoints still harbouring very high copy numbers of the ecDNA region (Fig. 3d). While CT scoring on a single cell level is less robust than at clone level (Supplementary Fig. 8a), cells with so few breakpoints on chromosome 2 are unlikely to be chromothriptic in this location. This could be explained by the presence of cells carrying the ecDNA, but having lost the CT chromosome. The presence of cells carrying only the ecDNAs but without the derivative CT chromosome suggests that the ecDNAs themselves may possibly provide a stronger selective advantage. In tumours with two or more ecDNAs originating from distinct chromosomes, the amounts of ecDNAs generated from different loci were tightly correlated only in a minority of cases, with most nuclei characterised by the presence or absence of each individual ecDNA (Supplementary Fig. 10a, b). We detected substantial heterogeneity across clones but also across cells within clones in the number of ecDNAs, suggesting that the number of ecDNAs is not directly linked with the copy-number profiles, or subclonal CT status, on this chromosome (Fig. 3e, Supplementary Fig. 10b, d). Essentially, this shows that ecD-NAs further add to the intra-tumour genetic heterogeneity. Cells with extreme levels of ecDNAs were rare and usually did not cluster with any other cell from the same tumour, suggesting that above a given threshold, the number of copies might not further increase the selective advantage for clonal expansion (Supplementary Fig. 10a). It is

conceivable that within a certain range a given oncogene may provide a selection advantage, but too high levels of specific oncogenes may become detrimental for the cell, as suggested by the concept of oncogene overdose[34]. The presence of ecDNAs was associated with higher RNA expression of oncogenes carried on the ecDNAs, with large variations in expression within tumours (Fig. 3f).

We experimentally validated the presence of ecDNAs by FISH in the tumour BT084, focusing on the CT event on chromosome 2 (Fig. 3g). For this purpose, we combined a FISH probe for *GLI2*, an oncogene carried by ecDNAs in this tumour, and a Xcyte 2 probe allowing us to visualise distinct regions of chromosome 2 with different fluorophores. The vast majority (close to 80%) of the analysed metaphases showed four copies of chromosome 2, with three copies of similar size and one shorter copy. We occasionally detected metaphases with three or five copies of chromosome 2, consistent with the single-cell CNV data showing intra-tumour heterogeneity regarding the presence of the CT chromosome across cells mentioned earlier.

To further assess the link between CT and ecDNAs in Sonic Hedgehog (SHH) medulloblastoma, we searched for ecDNAs in bulk WGS of SHH medulloblastoma ($n = 46$) and performed CT scoring (see "Methods"). Remarkably, medulloblastomas with ecDNAs showed a significantly higher CT prevalence (Fig. 3h), confirming the association suggested in a previous study in one medulloblastoma via inference of the ecDNA structure from bulk WGS[2]. In addition, we identified a significant correlation between the presence of ecDNAs and CT prevalence across nine tumour types, suggesting that CT is not only one way how ecDNAs are generated, but might be the major way (Fig. 3i). Together with our results from single-cell DNA-seq, these data show that CT and ecDNAs co-occur within the same tumours, while not necessarily being conserved within the same individual cells throughout tumour evolution.

## Integrating copy-number variation with transcriptional heterogeneity in medulloblastomas with chromothripsis

Beyond selective advantages provided by CT, we set out to characterise broader consequences of CT, in particular on the transcriptome of these tumours. While previous studies have considered scRNA-seq profiling to characterise medulloblastoma heterogeneity[35,36], the extent

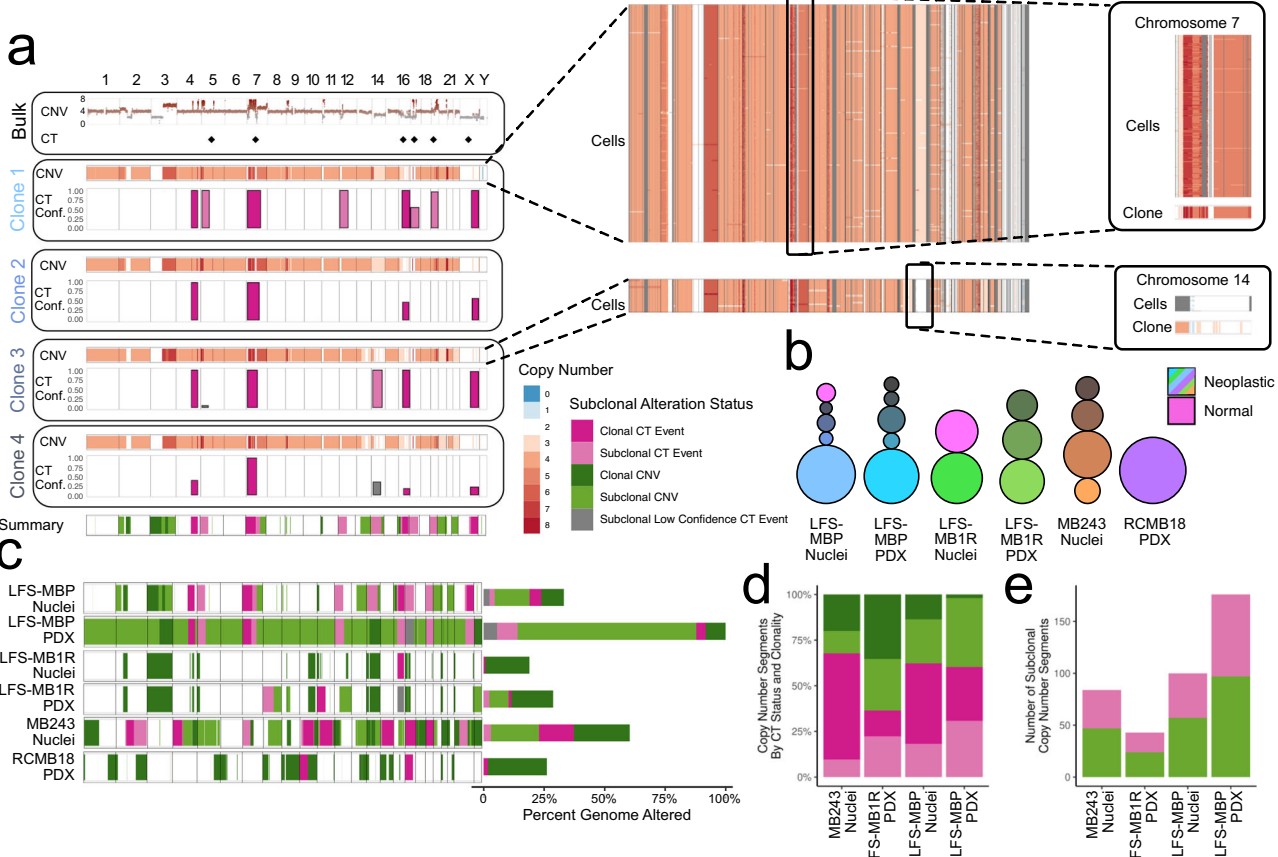

**Fig. 2 | Genetic heterogeneity in medulloblastomas with chromothripsis.** Throughout, clonal, subclonal or low confidence (<50% of bootstrap samples CT positive) CT events are coloured dark pink, light pink or grey respectively, and clonal and subclonal CNVs are coloured dark and light green respectively. **a** Copy number profiles and clonal substructure estimated using bulk WGS and scDNA-seq (LFS-MBP, primary tumour, *n* = 268 nuclei). From top to bottom: somatic copy number profiles from bulk DNA-seq (Bulk), with diamonds indicating chromosomes with evidence for chromothripsis (CT; estimated using ShatterSeek, see "Methods"); pseudobulk CNV profiles of 4 genetic clones (excluding a clone composed of normal cells) identified by hierarchical clustering of cell-level CNV profiles estimated from scDNA-seq. Barplots denote the fraction of bootstrap samples (B = 101) scored as CT positive ("Methods") below; the location and width of the bar correspond to the region overlapping CT positive windows (50 Mb windows evaluated at 20 kb resolution); a summary of the copy number alteration or CT status and clonality for each genomic region across clones. Right: Per-cell profiles for Clones 1 and 3, with a region of clonal CT on chromosome 7 highlighted for Clone 1, and region of subclonal CT on chromosome 14 highlighted for Clone 3. **b** Summary of the clonal structure identified in each scDNA-seq sample (*n* = 188 LFS-MBP PDX; *n* = 32 LFS-MB1R Nuclei; *n* = 38 LFS-MB1R PDX; *n* = 41 MB243 Nuclei; *n* = 27 RCMB18 PDX). Circles correspond to identified clones with the area of the circle proportional to the fraction of cells/nuclei assigned to this clone within each sample ("Methods"). Colour and numbering of clones is arbitrary, except clones identified as composed of normal cells coloured pink across samples. LFS-MBP PDX was derived from the tumour LFS-MBP (Nuclei); LFS-MB1R Nuclei and PDX are the matched relapse samples. **c** Genomic regions coloured by the presence of clonal or subclonal copy number alterations (from median sample ploidy; "Methods") across 6 samples with scDNA-seq. Regions of clonal, subclonal, or low confidence (grey) CT highlighted. Stacked bar plots on the right indicate the percentage of the genome altered by each of the 5 types of copy number alterations for each sample. **d** Percentage of individual CNV segments classified as clonal or subclonal CT associated, or clonal and subclonal non-CT associated. **e** Number of CNV segments with subclonal copy number changes that overlap with regions identified as CT versus segments outside CT regions. Source data are provided as a Source Data file.

to which a highly rearranged genome in CT medulloblastoma has transcriptional consequences is not well understood. We performed 10X single-cell (from PDX samples) and single-nuclei (from patient tumours) RNA-seq of the same samples subjected to single-cell DNA-seq (Fig. 4a, "Methods"), yielding 15,259 single-nuclei and 7241 single-cell (PDX) transcriptomes respectively (after QC; Supplementary Data 4; hereafter referred to as scSeq for both cells and nuclei). A joint embedding of cells across the entire dataset revealed, as expected, substantial heterogeneity between patients, tumours and their corresponding PDX models (Supplementary Fig. 11). Thus, we conducted clustering in each sample, followed by annotation using literature-derived[36–38] marker genes (Supplementary Data 5). This approach identified (Fig. 4b–e) three major malignant cell states (transcriptionally close to granule neuron progenitors, as expected, Supplementary Fig. 12a), which were detected both in nuclei and PDX, characterised by: SHH signalling activity (e.g., *GLI2*), proliferation (e.g.,

*MKI67*, *TOP2A*) and neuronal development and differentiation (e.g., *RBFOX3*, *NEUROD1*). Motivated by prior work identifying these three malignant cell states also in non-CT SHH medulloblastomas[35], we projected these single-cell profiles into an existing single-cell reference atlas of non-LFS SHH MBs (using ingest; "Methods"). This integration confirmed the transcriptional similarity between the CT tumour samples and the non-LFS SHH MBs, where samples matched primarily with the SHH group (Supplementary Fig. 12b, c), which is in line with classification results obtained from DNA methylation (Supplementary Fig. 12d). Collectively, these results indicate that, despite pronounced genomic differences between CT and non-CT medulloblastomas from the same molecular subgroup, the cellular programmes are qualitatively shared between both groups.

To dissect more subtle transcriptional differences between CT and non-CT medulloblastomas, we leveraged three existing bulk RNA-seq data resources (46 fresh-frozen[39] and 173 FFPE[40,41]

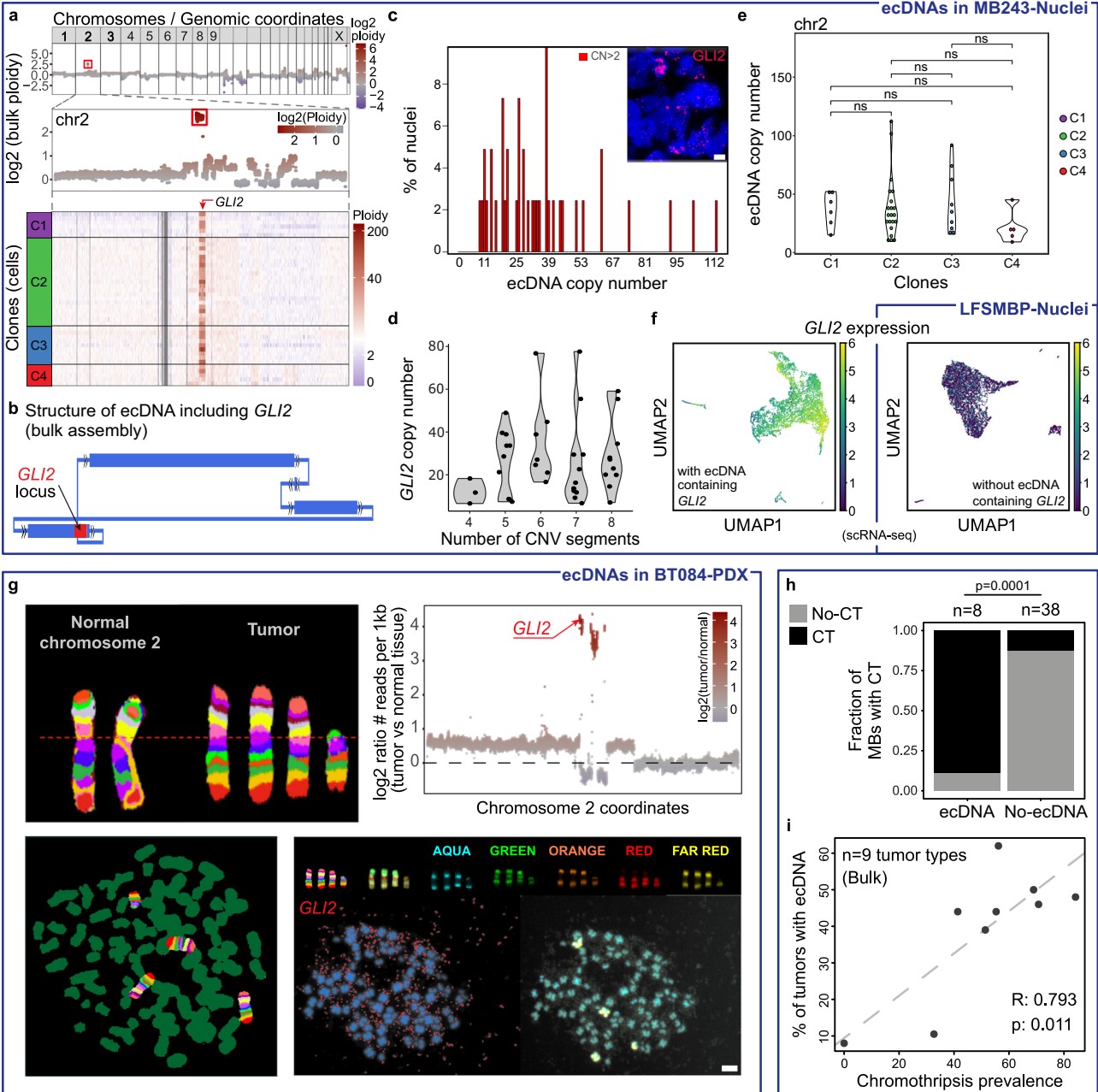

**Fig. 3 | Chromothripsis is a major event for the formation of ecDNAs. a** Copy number plots for ecDNA carrying the *GLI2* oncogene (whole-genome view and zoom on chromosome 2) generated by a CT event detected in the MB243-Nuclei sample (shown in panels **a–f**). **b** Structure of the ecDNA carrying *GLI2* confirmed by bulk WGS assembly (derived using AmpliconArchitect; "Methods"). Blue rectangles show genomic segments. Arrows denote the orientation of a segment from lower to higher coordinates. **c** scDNA-seq of tumour nuclei shows high copy-numbers of the segments included in the ecDNA (the number of copies of the *GLI2* locus located on the ecDNAs is used as a proxy for the number of ecDNA copies). Most tumour cells carry 10 to 40 copies of the ecDNA, with more than 100 copies per cell in extreme cases. Scale bar, 5 µm. **d** Relationship between CT on chromosome 2 (using the number of CNV segments on Chr. 2 per cell as a proxy for chromothripsis) and the number of copies of ecDNAs. Subsets of tumour cells carry both the CT chromosome and ecDNAs while other cells keep the ecDNA but may have lost the CT

chromosome. **e** Number of copies of ecDNAs per clone (using the number of copies of *GLI2* as a proxy). Significance is displayed from Bonferroni adjusted *p* values. **f** Tumour cells with ecDNAs including *GLI2* show a high expression of this oncogene (left, sample MB243-Nuclei; right, *GLI2* expression in sample LFS-MBP-Nuclei for comparison with a sample without ecDNA including *GLI2*). **g** FISH validation of ecDNAs carrying *GLI2* likely generated by CT on chromosome 2 (BT084-PDX sample). In addition to the *GLI2* probe (red), we used a multicolour probe for chromosome 2. Scale bar, 10 µm. **h** Sonic Hedgehog medulloblastomas with ecD-NAs have a significantly higher CT prevalence (*n* = 46; two-sided Fisher exact test). **i** CT is significantly linked with the presence of ecDNAs across nine tumour types (colon cancer, haematological malignancies, prostate cancer, breast cancer, ovarian cancer, melanoma, renal cancer, lung cancer and glioblastoma, two-sided pearson correlation test shown, reanalysis from[3,77]). R, Pearson correlation coefficient. Source data are provided as a Source Data file.

medulloblastomas with and without CT; see "Methods"; Supplementary Fig. 12e–h, Supplementary Data 6). Differential expression analysis between CT and non-CT SHH medulloblastomas identified a union of 916 differentially expressed genes across both FF and FFPE samples

(FDR < 0.05, Log2FC < −1 | Log2FC > 1, Benjamini-Hochberg adjusted, two-sided Wald test, accounting for SHH subgroups, "Methods"). The intersection of both gene lists yielded 4 genes which were up-regulated, (*MKI67IP*, *GLI2*, *CLASP1* and *TSN*), all of which are reported to be

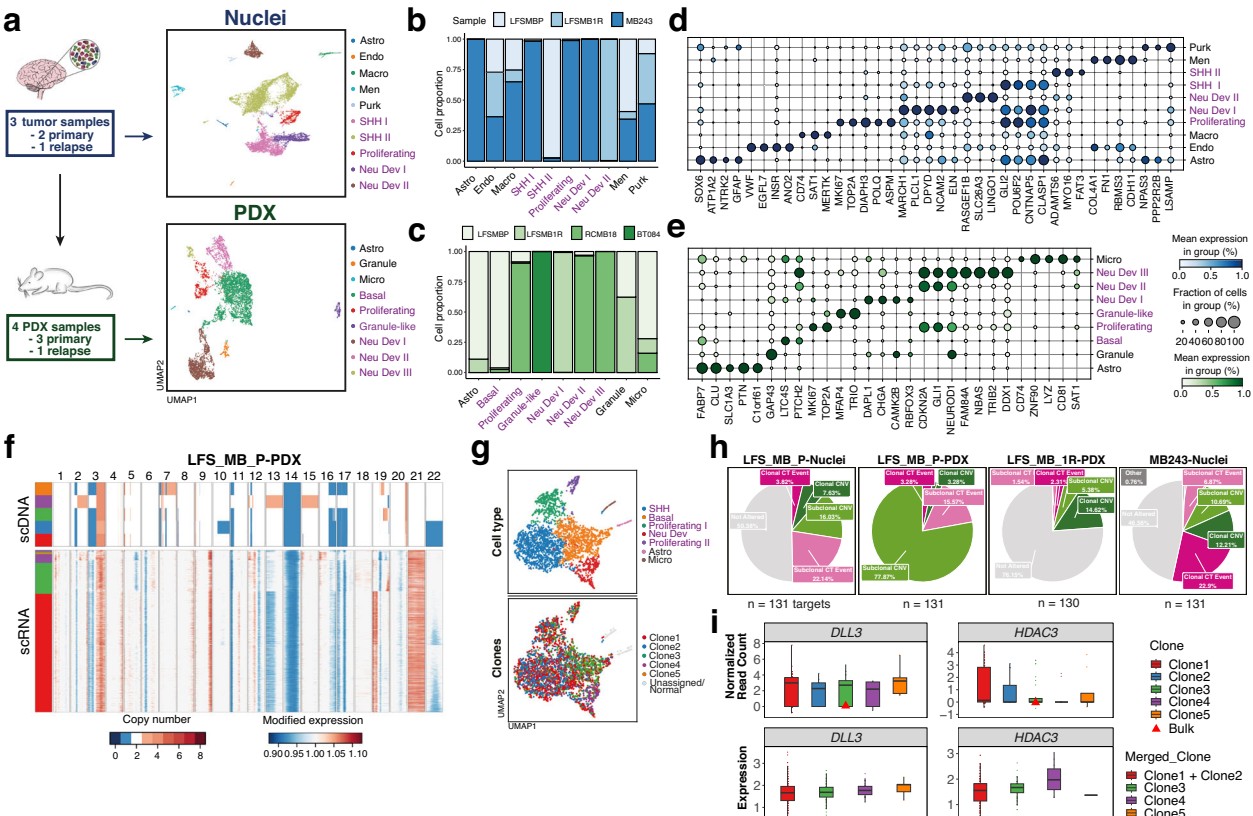

**Fig. 4 | Large-scale genomic alterations drive transcriptional heterogeneity in medulloblastoma with chromothripsis. a** Overview of the experimental procedure to generate the single-cell and -nuclei RNA-seq data for the 7 samples in this study. Top: single-nuclei RNA-sequencing (blue, tumours). Bottom: single-cell RNA-sequencing (green, PDX). UMAP embedding for tumour samples ($n = 3$), profiled using single-nuclei RNA-seq (left, 15,259 cells) and PDX ($n = 4$) profiled using single-cell RNA-seq (right, 7241 cells). Cell types annotated using literature-derived marker genes indicated in distinct colours. The Macro cell type contains both macrophage and microglia cells. Malignant cell types highlighted in purple; non-malignant cell types shown in black font. **b, c** Stacked bar plots, displaying the relative prevalence of individual cell types across samples, for tumours (**b**) and PDX models (**c**). Colours highlight distinct sample contributions. **d, e** Dotplot displaying the expression level and prevalence of marker genes (x-axis) across the cell populations identified (y-axis) as shown in (**a, b**). Dot size denotes the fraction of cells expressing the respective marker gene; dot colour depicts the relative expression level. **f** Results from the clone alignment of scDNA- and scRNA-seq profiles for LFS-MBP PDX. Top: Heatmap displaying the copy number profiles derived from scDNA, with colour corresponding to copy number (scale [0,8]; larger values clipped). Bottom: Heatmap showing relative CNV profiles estimated using inferCNV[42], with colour

corresponding to modified expression (scale centred at 1; diploid state). Colour bar on the right side indicates uncertainty measures for the assignment of copy number clones (see "Methods"). **g** UMAP embedding depicting 3629 cells from scRNA-seq after QC for LFS-MBP PDX. Cells are coloured according to their assigned cell type identity (top) or clone from scDNA (bottom). Phenotypically normal cells as well as cells that did not meet the assignment confidence were excluded and are marked as unassigned or normal cells. **h** Pie charts showing the fraction of druggable targets affected by distinct genomic alterations as highlighted in Fig. 1 across all aligned samples. **i** Boxplots showing the copy number status (top; normalised read counts multiplied by cell ploidy) and expression (bottom; ln[CP10K + 1] values) of *DLL3* and *HDAC3*, as examples of druggable targets, in clones from scDNA- and scRNA-seq for LFS-MBP PDX. Clone1 and Clone2 have been merged because their genomic profiles are too close to distinguish them accurately based on scRNA-seq (see "Methods"). Each point corresponds to a single cell, with corresponding cell numbers in each clone in Supplementary Data 7. Tukey boxplots are displayed, centred at the median, with hinges at 25th and 75th percentile, and whiskers extending to 1.5 the IQR. All outliers are plotted as individual points. Source data are provided as a Source Data file.

important for chromosomal translocations, SHH signalling, or MYC targets. Conversely, 14 genes were significantly down-regulated, including *AHNAK, LRP1, CDKL5*, and *HIST1H3J*. Altogether, the combined transcriptome analysis hinted towards strikingly subtle differences between CT and non-CT tumours, given the substantial difference in aggressiveness of CT as compared to non-CT medulloblastomas.

Next, to more explicitly study the effect of CT-induced CNVs on single-cell transcriptomes, we set out to integrate scDNA-seq and scRNA-seq profiles at the level of the subclones. Briefly, we used broad-scale genome-wide CNV profiles estimated from scRNA-seq (using inferCNV[42], "Methods", Supplementary Fig. 13, 14) to align individual cells to the most likely clone of origin (Supplementary Fig. 13, 14). While this approach was able to confidently assign at least 20 cells to each scDNA-seq clone (alignment confidence estimated using permutations; "Methods"), a large proportion of RNA cells did not have a

unique match to a scDNA-seq clone. To increase the fraction of cells with well-defined mappings, we collapsed clones with ambiguous assignments. This approach allowed us to align RNA cells to between 2 and 4 clones per sample (11 clones in total, out of 16 before collapsing) which correspond to an scRNA-seq assignment rate of 62 and 98% of scRNA-seq cells (Supplementary Data 7, Supplementary Fig. 15). For example, out of 5 scDNA clones in LFS-MB-PDX, cells from scRNA could be assigned to 4 distinct genomic groups, reflecting that major CNV events could be detected in both modalities (Fig. 4f). Notably, even though we observed an enrichment of certain cell types in each genetic clone, the dominant source of variation in the scRNA-seq data was cell type rather than clone (Fig. 4g, Supplementary Fig. 16). The clone labels also allowed us to identify molecular signatures of individual clones, such as differentially expressed genes and pathways (Supplementary Fig. 16 and Supplementary Data 8, FDR < 0.05, two-sided Wilcoxon rank sum test, Benjamini-Hochberg adjusted, one

clone versus all, "Methods"). This analysis revealed significant clone-specific expression of MYC targets in each sample and MTOR signalling in most (Supplementary Fig. 16, FDR < 0.05, Kolmogorov–Smirnov statistic, Benjamini-Hochberg adjusted). Of note, these molecular processes were in part consistent with but mostly distinct from the changes identified when assessing differential expression between CT and non-CT tumours using bulk RNA-seq (46 fresh frozen[39] and 173 FFPE[40,41], Supplementary Fig. 12; Supplementary Data 6).

We then set out to use this integrated resource to reassess known putative targets for personalised treatment in paediatric oncology[43]. Among the 131 known druggable targets that were expressed in our data, between 7 and 93% of these genes were overlapping with subclonal CT or subclonal CNVs (Fig. 4h), emphasising the importance of genetic heterogeneity for target selection. Leveraging the aligned transcriptome profiles at the level of clones, we observed the expected effect of genomic heterogeneity at the transcriptome level for some targets and identified exceptions where expression levels are not determined by copy number (Fig. 4i, Supplementary Fig. 17).

Taken together, the results from our integration suggest that the combined information from single-cell genomes and transcriptomes can reliably identify copy-number related pathway alterations, which are biologically relevant. Future studies focusing on larger cohorts of CT tumours will be crucial to further underline these findings and highlight the impact of copy number variation on the transcriptome, including on the expression of druggable targets that might seem clonal from bulk analyses only.

### Loss of chromosome 3p and *SETD2* deficiency as early events potentially facilitating chromothripsis

In addition to characterising pathways activated in specific clones, we leveraged the single-cell and bulk WGS data to identify putative early alterations that might contribute to CT occurrence. Previous studies indicated that inactivation of essential checkpoints likely facilitates CT and/or the survival of a cell after CT events[2,11,12]. We searched for early events potentially linked with CT. Loss of chromosome 17p (chr17p), carrying the wild-type *TP53* allele, was already known to be associated with CT in medulloblastoma patients with germline *TP53* mutations[2]. In agreement with this, rare non-tumour cells with a balanced profile (defined as non-tumour cells based on the absence of CNV), except a focal loss of the *TP53* locus, supported the loss of p53 as an early event in CT tumours (Supplementary Fig. 18a). In addition, we identified chromosome 3p (chr3p) loss as a clonal event linked with chr17p loss and CT (Fig. 5a). This was further supported by phylogenies reconstructed from deep bulk sequencing data and allele frequency analyses (Fig. 5b). Investigating this association in a cohort of 227 medulloblastomas, we found that loss of chr3p was highly significant when searching for genomic regions tightly linked with CT (Fig. 5c, two-sided Fisher exact test, $p < 10^{-5}$ and two-sided Chi-square test, $p < 10^{-8}$, respectively). Importantly, loss of chr3p was also significantly linked with CT in breast and lung cancer (two-sided Chi-square test, $p < 1.32 \times 10^{-4}$ for breast cancer and $p < 3.44 \times 10^{-10}$ for lung cancer), suggesting a potential pan-cancer relevance, beyond medulloblastoma (Supplementary Fig. 18b). To validate this association experimentally, we performed time-course analyses with primary cells from LFS patients. In these primary cultures (derived from patient skin biopsies and not subjected to induced immortalisation), we identified loss of both chromosomes 17p and 3p as early events linked with CT using WGS (Supplementary Fig. 18c).

To search for candidate genes on chr3p potentially preventing CT, we defined the minimally deleted region across bulk WGS data from 18 LFS medulloblastomas (Supplementary Fig. 19). We narrowed the list of candidates based on gene expression in LFS medulloblastomas, reported mutations and function. Among the evaluated genes, *SETD2* was a promising candidate, due to the known tumour suppressive role of the SETD2 methyltransferase, lost or mutated in various cancers,

and its importance for DNA replication, DNA repair and genome instability[44,45]. Medulloblastomas with chr3p loss displayed a lower *SETD2* expression based on single-cell and bulk RNA-seq (Fig. 5d–f). In addition, low *SETD2* expression was linked with significantly shorter overall survival in SHH medulloblastoma (Fig. 5g, two-sided log-rank test, $p < 0.018$, SHH alpha (SHH3) subgroup, which is the molecular subgroup to which most CT medulloblastomas belong). SHH medulloblastomas with CT displayed a lower protein expression as compared to SHH medulloblastomas without CT, as shown by immunohistochemistry (Fig. 5h, i).

As combined single-cell and bulk sequencing analyses identified *SETD2* as a promising candidate potentially preventing CT, we analysed the functional consequences of *SETD2* loss. To test for a potentially causal role of *SETD2* in CT, we used CRISPR/Cas9 to inactivate *SETD2* in p53 wild-type and p53-deficient neural stem cells, respectively (Supplementary Fig. 20a). CT has previously been linked with genome doubling[46], as well as with the formation of micronuclei[13,17], which are abnormal nuclear structures containing one or very few chromosomes. Our CRISPR/Cas9 experiments showed that, upon *SETD2* inactivation in a p53-deficient background, the formation of micronuclei significantly increased as compared to inactivation of *TP53* only (Fig. 6a, b, one-way Anova and Bonferroni multiple comparison tests, $p < 0.05$). In addition, as compared to wild-type cells, *TP53/SETD2* knock-out cells showed a significantly larger nuclear area (Fig. 6c, one-way Anova and Bonferroni multiple comparison tests, $p < 0.05$), a measure which is used as a surrogate marker for polyploidization[47]. Immunofluorescence analysis of the widely used DNA double-strand break marker γH2AX showed a significant increase in the levels of DNA double-strand breaks in *TP53/SETD2* knock-out cells as compared to wild-type neural stem cells (Fig. 6d, e, one-way Anova and Bonferroni multiple comparison tests, $p < 0.05$). Double stain for phosphorylated histone H3 and acetylated tubulin identified aberrant mitoses in *SETD2* and in *TP53/SETD2* knock-out cells, such as failure to congress at pro-metaphase, multipolar spindle formation and anaphase bridges (Fig. 6f, g). This is in agreement with CT being one consequence of bridge breakage[16]. We measured a significantly increased proliferation rate upon *TP53/SETD2* knock-out (Fig. 6h), indicating a selective advantage. Finally, strand-seq analysis showed significantly more structural variants including complex rearrangements in the knock-out cells (Fig. 6i, j, Supplementary Fig. 20b). Altogether, the functional consequences of the inactivation of *TP53* and *SETD2* in neural stem cells suggest a possible causative or permissive role for these two genes in the occurrence of CT in medulloblastoma.

## Discussion

Li-Fraumeni syndrome (LFS) medulloblastoma is a clinically challenging type of childhood brain tumour, where patients suffer from a dismal prognosis. These tumours are a canonical model of CT, an extreme phenomenon of genome instability, which is present in close to 100% of these medulloblastomas[2,26]. Hence, understanding the genomic heterogeneity and its consequences on the transcriptome are essential to identify targets for novel therapeutic strategies for this subgroup of patients.

In this study, we combined single-cell analyses of the genome and transcriptome together with bulk deep sequencing to provide a roadmap of alterations in CT medulloblastoma with *TP53* germline mutations. By combining bulk and single-cell DNA sequencing of matched tissue samples, we scored CT events at clonal resolution. This approach enabled us to shed light into the genomic heterogeneity at the level of CT chromosomes, cancer drivers as well as potentially druggable targets. In addition, we observed and experimentally validated an association between the abundance of ecDNA structures and CT, further increasing the bespoke heterogeneity, as ecDNAs can be a consequence of CT but also substrates of additional CT events. Comparisons between matched primary and relapse samples in patient

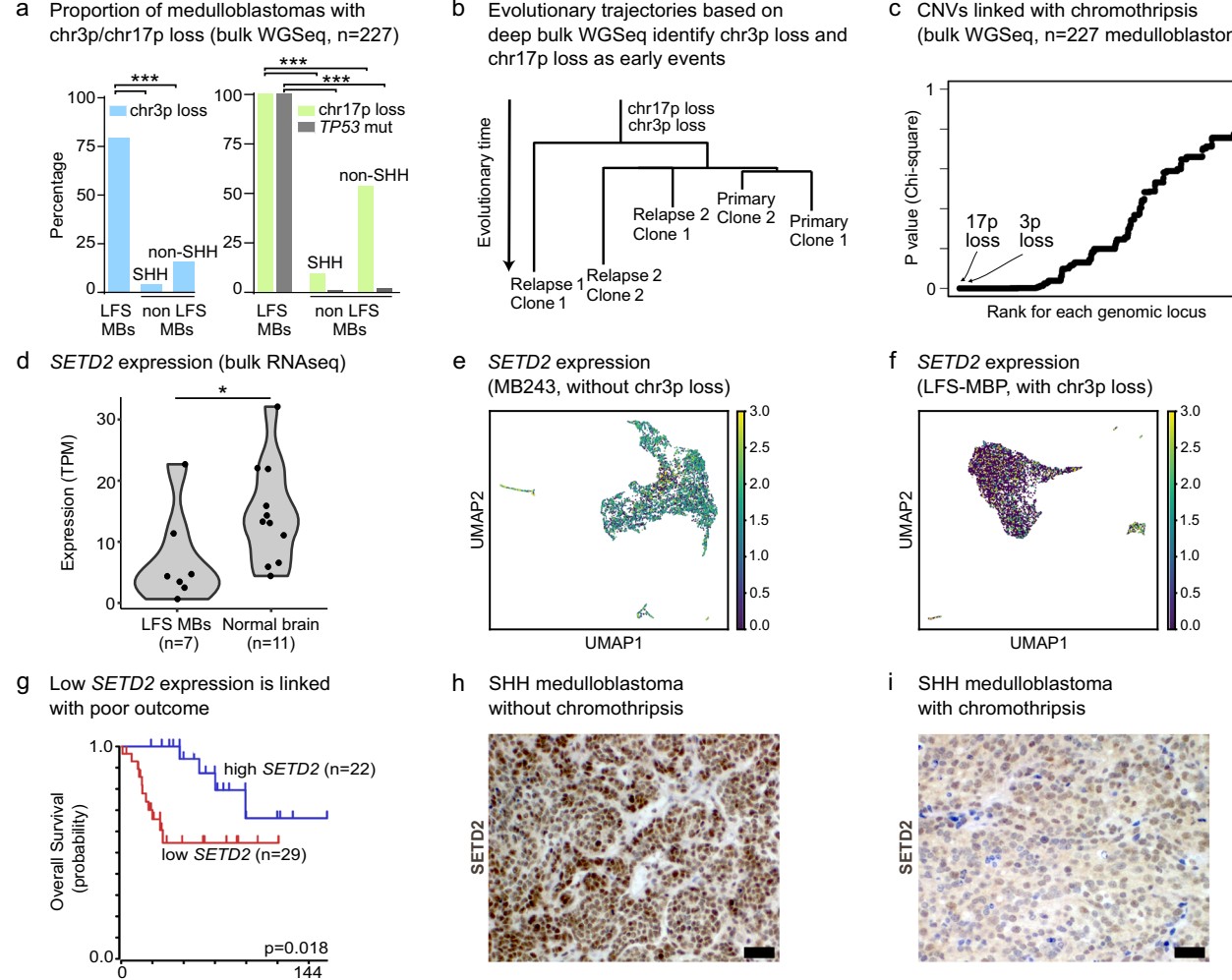

**Fig. 5 | Combining single-nuclei DNA-seq and bulk whole-genome sequencing identifies early events potentially facilitating chromothripsis. a** Proportion of medulloblastomas with chromosome 3p or 17p loss among tumours with CT (LFS medulloblastomas) and non-LFS medulloblastomas (*n* = 227). Two-sided Fisher exact tests were performed to compare the proportions of tumours with 3p or 17p loss between LFS medulloblastomas and non-LFS medulloblastomas (*p* < 0.00001). **b** Evolutionary trajectories based on deep WGSeq identify 17p loss and 3p loss as early events (longitudinal analysis of three matched tumour samples). **c** 17p loss and 3p loss are significantly linked with CT in medulloblastoma (bulk WGS, *n* = 227 medulloblastomas). **d** Loss of 3p leads to decreased *SETD2* expression (bulk RNA-

seq, *n* = 18, *p* = 0.0348). Statistical significance was tested using one-tailed *t*-test. **e, f** Loss of 3p leads to decreased *SETD2* expression (scRNA-seq). **g** Low *SETD2* expression is linked with poor survival in medulloblastoma (SHH alpha subtype, *n* = 53, enriched for p53-SHH medulloblastomas, log-rank test, Kaplan-Meier plot generated using the R2 database, see "Methods"). **h, i** Representative examples of medulloblastomas with or without CT showing low or high SETD2 protein expression, respectively. Immunohistochemistry analysis was performed in eight patient samples showing similar results (*n* = 4 medulloblastomas with CT; *n* = 4 medulloblastomas without CT). Scale bar, 50 μm. Source data are provided as a Source Data file.

tumours and PDX, on both genome and transcriptome, supported substantial heterogeneity with major implications for treatment. Importantly, our results also question the common view of CT as a single early event in tumour development, which goes along with limited intra-tumour heterogeneity.

We aimed at identifying putative early events in LFS tumour evolution. It has been unclear whether the inactivation of essential checkpoints such as p53 and others, may occur shortly before or after CT. Here, our results highlighted chr3p loss and *SETD2* inactivation as a potential early event facilitating CT occurrence. We experimentally underlined this observation utilising CRISPR/Cas9-mediated inactivation of *SETD2* in p53 wild-type and p53-deficient neural stem cells. In line with this, we detected rare non-tumour cells with *TP53* loss (potentially primed for CT), but no tumour clones with loss of chr17p and/or chr3p without CT. However, as such tumour cells are expected at a very low frequency, sequencing thousands of tumour cells would be necessary to detect such rare populations. To recapitulate the

sequence of events, we used a time-course experiment, culturing primary fibroblasts from early passages with stable copy-number profiles to late passages with spontaneous CT occurrence. Our findings validated chr17p loss and chr3p loss as early events correlated with CT. It will be important to understand why only specific cell types ultimately lead to CT tumours even though all cells in LFS patients harbour mutant *TP53*.

So far, the transcriptional consequences of CT in tumours have not been systematically investigated. Here, we leveraged single-cell and single-nuclei RNA-seq to analyse 7 samples from LFS medulloblastoma and PDX samples. Remarkably, we found a variety of malignant and non-malignant cell types, a subset of which were represented in the PDX samples. Furthermore, we observed three transcriptional programmes largely defined by (i) SHH genes, (ii) proliferation and (iii) genes implicated in neuronal development. These programmes were surprisingly consistent with programmes previously observed in non-CT SHH medulloblastoma[35]. Our analysis of bulk RNA-seq from CT and

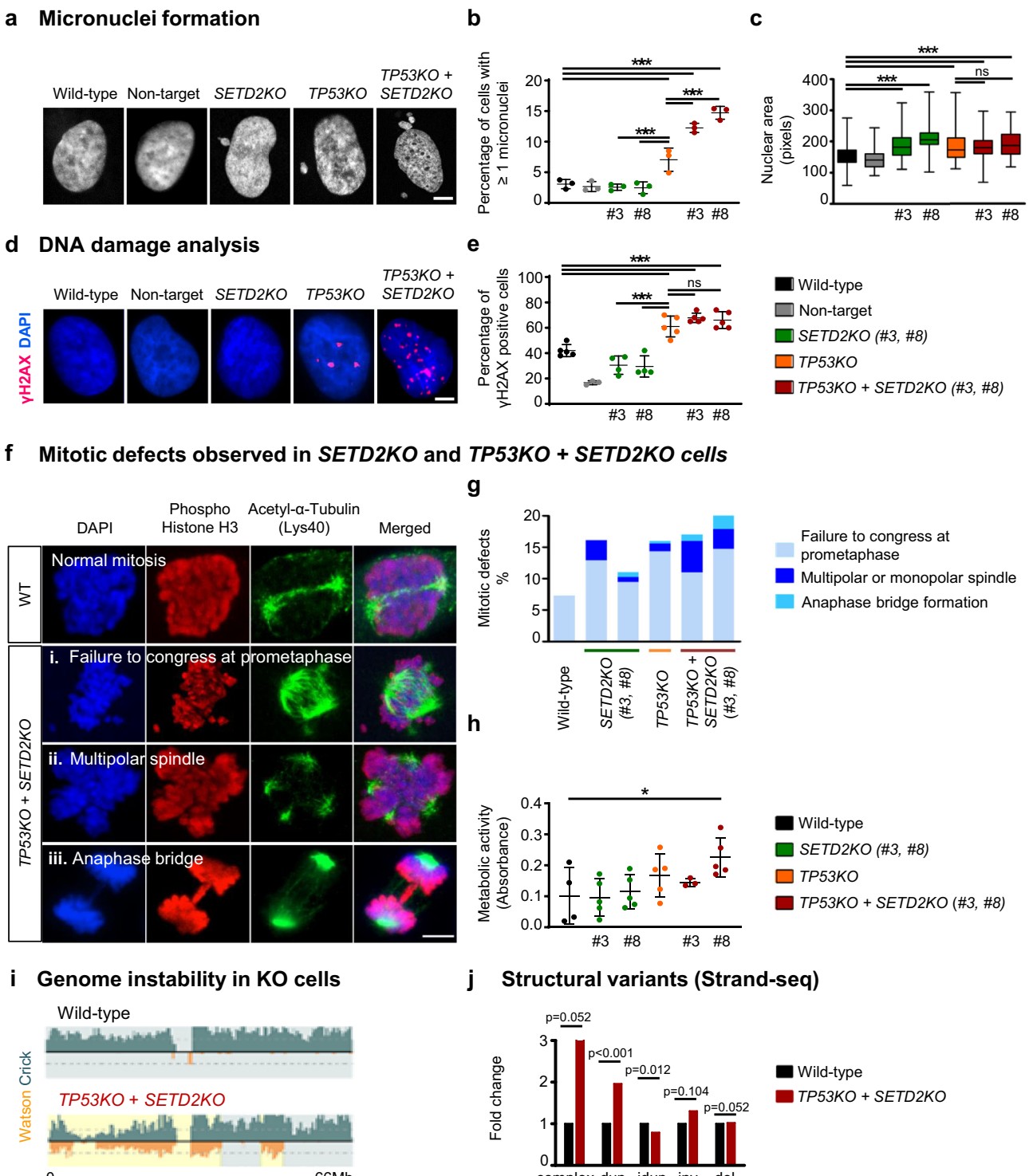

**a  Micronuclei formation**

**b** Percentage of cells with ≥ 1 micronuclei

**c** Nuclear area (pixels)

**d  DNA damage analysis**

**e** Percentage of γH2AX positive cells

- ■ Wild-type
- ■ Non-target
- ■ *SETD2KO (#3, #8)*
- ■ *TP53KO*
- ■ *TP53KO + SETD2KO (#3, #8)*

**f  Mitotic defects observed in *SETD2KO* and *TP53KO + SETD2KO cells***

DAPI | Phospho Histone H3 | Acetyl-α-Tubulin (Lys40) | Merged

WT — Normal mitosis

TP53KO + SETD2KO:
i. Failure to congress at prometaphase
ii. Multipolar spindle
iii. Anaphase bridge

**g** Mitotic defects %

- Failure to congress at prometaphase
- Multipolar or monopolar spindle
- Anaphase bridge formation

**h** Metabolic activity (Absorbance)

- ■ Wild-type
- ■ *SETD2KO (#3, #8)*
- ■ *TP53KO*
- ■ *TP53KO + SETD2KO (#3, #8)*

**i  Genome instability in KO cells**

Wild-type / TP53KO + SETD2KO; Watson Crick; 0 – 66Mb

**j  Structural variants (Strand-seq)**

Fold change: complex (p=0.052), dup (p<0.001), idup (p=0.012), inv (p=0.104), del (p=0.052)

- ■ Wild-type
- ■ *TP53KO + SETD2KO*

non-CT medulloblastomas emphasised differences in gene expression and activated pathways, in particular regarding MYC driven transcription, SHH signalling, and proliferation. However, future studies focusing on the origin of the aggressiveness of CT tumours will be needed in order to dissect the precise mechanisms explaining the poor outcome of patients with CT tumours.

Lastly, to link the genome and the transcriptome data, we demonstrated how copy number estimates allow for assigning single-cell transcriptomes to individual CNV clones. Even though this computational integration of CNV clones and scRNA-seq may not directly transfer to settings with less pronounced genomic aberrations, by

linking distinct transcriptional profiles to the identified tumour clones, we were able to highlight differentially activated pathways between clones, including but not limited to MYC and MTOR signalling. Differential activity of oncogenic signalling pathways has important implications in the context of drug response and treatment resistance.

This study does not come without limitations. A larger sample size would presumably be needed in order to identify commonalities between CT and the establishment of ecDNAs. In addition, although we investigated potential druggable targets in both scDNA and scRNA data, a larger number of matched primary tumours and relapse samples would be required to understand the influence of treatment on

**Fig. 6 | Inactivation of *SETD2* and *TP53* in neural stem cells leads to genome instability. a** Inactivation of *SETD2* in a p53 deficient background leads to the formation of micronuclei, aberrant nuclear structures linked with genome instability. Representative images based on three independent experiments are shown. Scale bar, 5 μm. **b** Quantification of micronuclei ($n$ = three biological replicates; mean ± SD; $p < 0.0001$). **c** Inactivation of *SETD2* leads to a larger nuclear area ($n$ = three biological replicates; $p < 0.0001$). The bounds of the box represent the interquartile range (25th–75th percentile), the central line marks the median, and the whiskers extend to the minimum and maximum values. **d, e** Inactivation of *SETD2* in a p53 deficient background leads to high levels of DNA double-strand breaks. Immunofluorescence analysis of γH2AX foci and quantification of γH2AX positive cells (Wild-type, $n$ = five biological replicates; Non-target, $n$ = three biological replicates; *SETD2KO* (#3, #8), $n$ = four biological replicates; *TP53KO*, $n$ = five biological replicates; *TP53KO + SETD2KO* (#3, #8), $n$ = five biological replicates; mean ± SD; $p < 0.0001$). Scale bar, 5 μm. **f, g** Inactivation of *SETD2* in a p53 deficient background leads to mitotic defects, as shown by immunofluorescence analysis of

Phospho Histone H3 and Acetyl-α-Tubulin. Scale bar, 5 μm. **h** Inactivation of *SETD2* in a p53 deficient background leads to increased proliferation rate. Metabolic activity results, indicating proliferation rate, are shown as absorbance values measured by MTT assay (Wild-type, $n$ = four biological replicates; *SETD2KO* (#3, #8), $n$ = five biological replicates; *TP53KO*, $n$ = five biological replicates; *TP53KO + SETD2KO* (#3), $n$ = three biological replicates; *TP53KO + SETD2KO* (#8), $n$ = five biological replicates; mean ± SD; $p = 0.0392$). **i, j** Strand-seq analysis of wild-type and knock-out cells. Quantification of structural variants was performed with MosaiCatcher. Beta regression and Bonferroni–Holm method for multiple comparisons were used to test for statistical significance in **b, e**. One-way ANOVA and Bonferroni multiple comparison tests were used to test for statistical significance in **c, h**. Wald test on the inactivation status explaining counts of observed events in a negative binomial GLM (inactivation status and intercept) fit for each event type independently was used to assess significance in (**j**). Two sided $p$ values are reported in (**b, e, j**). Source data are available as a Source Data file.

the intra-tumour heterogeneity in LFS medulloblastoma. Following this further, we would envision a larger cohort of CT and non-CT medulloblastomas being essential to get insights into the origin of the poor prognosis and aggressiveness of these tumours.

Tumours in LFS patients constitute a paradigm for the understanding of CT. Our work focusing on this group of patients can provide a roadmap from where the findings may be extended to different contexts, as the link between CT and *TP53* mutations also holds true outside the context of constitutive defects (e.g. in prostate cancer[48] or breast cancer[49]). In the future, a more refined single-cell landscape of CT tumours will be needed to further confirm and increase the understanding of the genomic heterogeneity, diversity of cell types and active transcriptional programmes. Unravelling the extent of genomic heterogeneity will be necessary to detect actionable targets, determine the evolutionary history and defeat the evolutionary capacity of tumour cells with high genome instability.

## Methods
### Experimental methods
**Sample cohort, DNA extraction and whole genome sequencing.** Human clinical samples and data were collected after receiving written informed consent in accordance with the Declaration of Helsinki and approval by the ethics committee of the Medical Faculty of Heidelberg University. Although sex was not used as an inclusion criteria for this study, all samples analysed were male. All tumours used for bulk sequencing had a tumour cell content confirmed by neuropathological evaluation of the hematoxylin and eosin stainings. DNA was extracted from frozen tissue using Qiagen kits. Purified DNA was quantified using the Qubit Broad Range double-stranded DNA assay (Life Technologies, Carlsbad, CA, USA). Genomic DNA was sheared using an S2 Ultrasonicator (Covaris, Woburn, MA, USA). Whole-genome sequencing and library preparations for tumours and matched germline controls were performed according to the manufacturer's instructions (Illumina, San Diego, CA, USA or NEBNext, NEB). The quality of the libraries was assessed using a Bioanalyzer (Agilent, Stockport, UK). Sequencing was performed using the Illumina X Ten platform.

### Sample collection and establishment of patient derived xenografts
All animal experiments were performed in accordance with the ethical and legal regulations for animal welfare and approved by the governmental council (Regierungspräsidium Karlsruhe, Germany). Orthotopic patient-derived xenografts were established in 6-10-week-old female immune-compromised mice (NSG, NOD.*Cg-Prkdc^{scid}Il2rg^{tm1Wjl}*), obtained from the DKFZ animal breeding facility. Patient-derived tumour cells were injected into the cerebellum, as

described previously[50] and outlined here. Before starting the orthotopic brain injection procedure, the animal was anaesthetised using inhaled isoflurane (2.5 Vol %) and placed in the mouse stereotaxic frame. Bepanthen (Bayer Vital GmbH, #1578675) was applied to both eyes as a lubricant. An incision of approx. 1 cm was made on top of the head, in the area between the ears, using a disposable scalpel (#NC9999403). Sterilised forceps (Fine Science Tools, #91100-12) were used to keep the skin on the side, exposing the skull. To clean the area of the exposed skull from any blood or connective tissue, clean cotton-tipped swabs were applied. An 18 G needle (CHIRANA T. Injecta, # CH18112) was used to make a burr hole in the cerebellum. The location of the burr hole, through which the tumour cells will be injected, was determined from the lambda (approx. 2 mm towards the back of the brain and 1 mm to the left). Once the hole was made, 4 μL of the cell suspension was taken in NanoFil 10 μL syringe (World Precision Instruments) with 26 G beveled NanoFil needle (World Precision Instruments #NF26BV-2). The syringe was placed in the syringe holder of the stereotaxic frame and positioned on top of the burr hole. Once the tip of the injection needle was in the burr hole, the needle was inserted 2 mm down in the cerebellum. The cell suspension was then deposited in the cerebellum by pressing the injection run key on the control instrument. Once the injection was completed, the syringe with needle was slowly moved up and removed from the stereotaxic frame. The incision was closed by joining the skin together with the forceps and bonding it with surgical glue (Braun, #9381104). On the completion of incision closure, the animal was placed in a clean recovery cage until it fully woke up from anaesthesia. The criteria for terminating animal experiments were strictly adhered to, involving regular monitoring for the following symptoms: skull bulging, ataxia (impaired balance and movement indicative of brain damage), hyperactivity, central or peripheral paralysis, reduced movement, lack of food or water intake, behavioural signs of pain and weight loss exceeding 20%. Housing conditions for the mice included a 12-hour light/12-hour dark cycle, an ambient temperature of 20–24 °C and relative humidity of 45–65%.

### Nuclei isolation from tumour tissue
Frozen tumour tissue was used for nuclei isolation. Tissue was cut using a scalpel with 1 mL of lysis buffer. After adding 4 mL of lysis buffer, the suspension was transferred to a glass douncer. A total of 20 strokes were used to dounce the suspension on ice with two different types of pestles. The entire suspension was then filtered with a 100 μm filter and then with a 40 μm filter into precooled falcon tubes. After centrifugation for 5 min at 555 g at 4 °C, the supernatant was removed and the pellet was resuspended in 5 mL of lysis buffer without Triton-X and DTT. This centrifugation step and the resuspension were carried out 3 times in total. The final pellet was then resuspended in

1 mL of nuclei storage buffer in 1.5 mL LoBind Eppendorf tubes for further analysis.

## FACS

Viably frozen patient-derived xenograft (PDX) cells were thawed in a 37 °C water bath and suspended in high-purity PBS supplemented with 10% foetal calf serum. The cells were then washed twice and centrifuged at 1000 rpm for 5 min at 4 °C. Single-cell sorting of the PDX suspensions was performed using a BD FACSAria II flow cytometer. Propidium iodide (PI), at a final concentration of 1 μg/mL, was added to distinguish and exclude dead cells from the sorting process. Contaminating mouse cells were gated out based on distinct forward scatter (FSC) and side scatter (SSC) profiles, which reflect the cell size and internal complexity/granularity of the cells.

## 10X single-cell RNA-sequencing library preparation

The single cell suspensions of PDX cells or nuclei from frozen tissue specimens were loaded on a 10x Chromium Single Cell instrument (10x Genomics, California) to generate single-cell Gel Bead-In-Emulsions (GEMs). Single-cell RNA-seq libraries were prepared using Chromium Single-Cell 5′ Library and Gel Bead Kit (PN1000014, 10x Genomics). Barcoding and cDNA synthesis were performed according to the manufacturer's instructions. In short, GEMs were created where all cDNA from one cell shared a common 10x barcode. GEMs were then incubated at RT and cleaned up using Dynabeads. After post GEM-RT clean-up, full length cDNA was generated by PCR with a total of 14 cycles for library construction. The cDNA libraries were constructed using the 10x ChromiumTM Single Cell 5′ Library Kit according to the manufacturer's protocol. In brief, the major steps for the library preparation included (i) Target enrichment from cDNA, (ii) Enriched library construction, (iii) 5′ Gene expression library construction and QC. For final library QC, 1 μL of the sample was diluted 1:10 and ran on the Agilent Bioanalyzer High Sensitivity chip.

## 10X single-cell DNA-sequencing library preparation

The single-cell suspensions from tumour nuclei or PDX cell samples were processed using the Chromium Single-Cell CNV Kit (10× Genomics) according to the manufacturer's protocol. In brief, using cell bead polymer, single cells or nuclei were partitioned in a hydrogel matrix on Chromium Chip C. Once the cell beads were encapsulated and incubated, they were subjected to enzymatic and chemical treatment. This lysed the encapsulated cells and denatured the gDNA in the cell bead, to make it accessible for further amplification and barcoding. A second encapsulation was performed to achieve single cell resolution by co-encapsulating a single cell bead and a single barcoded gel bead to generate GEMs. Immediately after GEM generation the gel bead and cell bead were dissolved. Oligonucleotides containing standard Illumina adaptors and 10x barcoded fragments were then amplified with 14 PCR cycles during two-step isothermal incubation. After incubation the GEMs were broken and pooled 10x barcoded fragments were recovered. For final sequencing library QC, 1ul of the sample was diluted 1:10 and ran on the Agilent Bioanalyzer High Sensitivity chip. Although the experiment was performed, the library for the BT084-PDX sample did not pass the quality control steps and hence was not included in this study.

## Sequencing of single-cell DNA and RNA libraries

Single-cell libraries were sequenced on the Illumina NextSeq and NovaSeq (paired-end sequencing).

## Fluorescence in situ hybridisation (FISH)

Nick translation was carried out for BAC clones obtained from Source Bioscience (*GLI2*, clone RP11 297J22). The probes were indirectly labelled via Nick translation. Detection was done with a rhodamine-labelled probe and a FITC-labelled probe. Pre-treatment of slides, hybridisation, post-hybridisation processing and signal detection were performed using standard protocol. Samples showing sufficient FISH efficiency (>90% nuclei with signals) were evaluated. Signals were scored in, at least, 100 non-overlapping metaphases. After the *GLI2* FISH to detect double-minute chromosomes, the coverslip was removed, and metaphase spreads were washed. Denaturation was performed to remove the signal from the *GLI2* probe and hybridisation was done using the multicolour XCyte 2 probe from Metasystems according to the manufacturer's instructions.

## Cell culture

Neural stem cells (human iPSC derived NSCs, kindly provided by Dr. Daniel Haag) were cultured in matrigel (Corning, 356230) coated 6-well plates, in NeuroCult NS-A proliferation media kit (Stemcell Technologies, #05751) supplemented with 40 ng/mL EGF (Sigma, #E4127), 40 ng/mL FGF (Preprotech, #GMP100-18 B), 10 ng/mL hLIF (Millipore, #LIF1010) and 10 μM Rock inhibitor Y-27632 (Enzo, #ALX-270-333-M001). Prior to the experiment, the cells were tested negative for mycoplasma contamination. Cell line identity was not authenticated as the lines are not commercially available.

## CRISPR-Cas targeted gene disruption

Guide RNAs for *TP53* and *SETD2* were constructed and cloned into lentiCRISPRv2 (Addgene, 52961) according to the original online protocol of the Zhang lab (http://www.genome-engineering.org/crispr/wp-content/uploads/2014/05/CRISPR-Reagent-Description-Rev20140509.pdf). Following genes were targeted:

*TP53* (gRNA2:CGACCAGCAGCTCCTACACCGG) *SETD2* (gRNA3:AATGAACTGGGATTCCGACG, and gRNA8:GGACTGTGAACGGACAACTG).

Lentiviral production was conducted using the recommendation of The RNAi Consortium. First, for each lentiCRISPRv2 plasmid to be transfected, $4 \times 10^6$ million HEK293T cells (below passage 10) were seeded into two 10 cm culture dishes in 6 mL medium (DMEM with 10% FCS). After 24 h, packaging plasmids psPAX2 and pMD2.G and the lentiCRISPRv2 plasmid containing the construct of interest were co-transfected into the HEK293T cells. The steps of co-transfection were the following: (1) Medium in each dish was replaced with 6 mL of fresh medium (DMEM with 10% FCS); (2) 600 μL Opti-MEM medium (Thermo Scientific, #31985062) was pipetted in a sterile 1.5 mL Eppendorf; (3) 30 μL TransIT®-LT1 transfection reagent (VWR, #731-0027) was added to 600 μL Opti-MEM (without mixing) and incubated for 5 min at room temperature; (4) 4 μg of each packing plasmid and 8 μg of the lentiCRISPRv2 plasmid were added to the Opti-MEM containing TransIT®-LT1; (5) the Eppendorf was closed and gently inverted four times to mix the reagents; (6) the solution was incubated for 20 min at room temperature; (7) 300 μL of the solution was added dropwise in each dish with HEK293T cells; (8) the cells were incubated for 72 h under standard cell culture conditions (37 °C, 5% $CO_2$). The whole procedure of the lentiviral production was conducted in the S2 laboratory. After 72 h of incubation, the supernatant from each dish was removed and filtered through a 0.45 μm filter to avoid contamination with cell debris. To concentrate the virus, the resulting virus containing solution was ultracentrifuged using a SW41 swing-out rotor in a L8-M ultracentrifuge at 25,000 rpm for 90 min at 4 °C. The supernatant was decanted and the virus pellet was resuspended in 100 μL sterile PBS. The concentrated virus was divided in 10 μL aliquots and stored at −80 °C until further use. Neural stem cells were cultured in matrigel coated 6-well plates as described above. Transduction of wild-type and *TP53KO* neural stem cells was done by adding 20 μL of concentrated virus particles to the cells for 24 h, after which the cells were maintained under selection with 2 μg/mL puromycin for 2–4 weeks. For CRISPR-mediated

disruption of *TP53*, an additional selection for functional knockout was done using 20 μM nutlin treatment. After selection for the stable lines, cell lysates were made for western blotting and cells were grown for further experiments.

## Western blotting

For western blot experiments, NSCs were detached with accutase (Sigma, #A6964), collected in media and washed three times with ice-cold PBS. The pellet was resuspended and incubated for 10 min on ice in RIPA buffer containing Complete™, EDTA-free Protease Inhibitor Cocktail (Sigma, #4693159001) and benzonase (Millipore, #71205-3). Protein concentration was estimated using BCA assay. To prepare samples for denaturing gel electrophoresis, samples were mixed with NuPAGE™ LDS Sample Buffer (4x) (Invitrogen, #NP0007), NuPAGE™ Sample Reducing Agent (10x) (Invitrogen, #NP0009) and deionized water. A total amount of 30 μg protein was loaded per lane of NuPAGE Tris-Acetate Protein Gel 3-8% (Life Technologies, #EA0375BOX) and separated in NuPAGE Tris-Acetate SDS Running Buffer (Invitrogen, #LA0041) for 1 h at 150 V constant. Immunoblotting was done on PVDF membranes in a tank blot system, using a borate-based buffer system (25 mM sodium borate, 1 mM EDTA, pH 8.8). Membranes were blocked with 5% milk powder in TBST for 1 h and probed with SETD2 (E4W8Q) rabbit mAb (Cell Signalling, #80290, Lot#1) 1:1000 overnight at 4 °C with agitation, TP53 (DO-1) mouse mAb (Santa Cruz, #sc-126) 1:500 for 1 h at RT, GAPDH (6C5) mouse mAb (Sigma-Aldrich, #CB1001) 1:2000 for 3 h at RT and H3K36me3 rabbit pAb (Abcam, #ab9050) 1:400 for 1 h at RT. Membranes were washed with TBST and incubated for 30 min with HRP coupled secondary anti-mouse or anti-rabbit antibodies (Dianova, #115-035- 003 and #211-032-171) 1:3000 (for H3K36me3) and 1:5000 for others. After washing, detection was done using enhanced chemiluminescence and images were recorded with the Bio-rad Imaging System (LI-COR Biotechnology).

## H&E stain and immunohistochemistry

Hematoxylin and eosin (H&E) staining and immunohistochemistry were performed on 4 μm formalin-fixed paraffin-embedded sections. Sections were deparaffinized, antigen retrieval was performed in 10 mM citrate buffer pH 6.0 for 40 min and sections were cooled down to room temperature. H&E stain was evaluated by a neuropathologist. For immunohistochemistry, rabbit polyclonal SETD2 antibody (Atlas Antibodies, #HPA042451) was used at 1:500 with the DCS SuperVision 2 HRP Kit.

## Immunofluorescence analysis of γH2AX in neural stem cells

Cells grown on coverslips were washed with PBS and incubated 15 min in 4% formaldehyde (formalin solution buffered at pH 6.8, Merck). Cells were washed once with 50 mM ammonium chloride (Carl Roth) and twice with PBS before permeabilization with 0.1% triton (Triton X-100, Gerbu Biotechnik). Blocking was done with 10% donkey serum (Merck). Primary antibody anti-gamma H2A.X (phospho S139) rabbit pAb (Abcam, #ab11174) diluted 1:200 in 10% serum was added and incubated overnight at 4 °C. After washing, coverslips were incubated with secondary antibody, washed in PBS, then in water and ethanol. Coverslips were mounted with DAPI Fluoromount (Southern Biotech, #0100−020).

## Quantification of γH2AX foci, micronuclei and nuclear area

Quantification was performed by visual examination under Axio Zeiss Imager.M2 microscope. The number of γH2AX positive NSCs was analysed by scoring at least 100 cells per line in five independent biological replicates. The number of cells containing micronuclei was analysed in at least 600 cells per line in three independent replicates. Nuclear area was calculated using a macro and scored in at least 100 cells per line in three independent replicates.

## Confocal imaging of mitotic errors

For acetyl-α-tubulin and phospho-histone 3 immunostaining, cells were seeded onto coverslips in a 6 well plate. The coverslips were then fixed for 20 min with 4% PFA. Next, a blocking buffer (1x PBS, 5% normal goat serum, 0.3% Triton X-100) was prepared and added to the coverslips for 1 h at room temperature. The blocking buffer was then removed and Acetyl-α-Tubulin (Lys40) (D20G3) XP rabbit mAb (Cell Signalling, #5335, Lot#5) and Phospho-Histone H3 (Ser10) (6G3) mouse mAb (Cell Signalling, #9706, Lot#10) primary antibodies were diluted to 1:400 and 1:200 respectively in an antibody dilution buffer (1x PBS, 1% BSA, 0.3% Triton X-100) and both were added simultaneously to each coverslip. Coverslips were incubated with the primary antibody overnight at 4 °C. Then cover slips were washed thrice for 5 min in 1X PBS. Subsequently goat anti-mouse and anti-rabbit secondary antibodies were diluted to 1:500 in the antibody dilution buffer and both were added simultaneously to each coverslip. Coverslips were incubated for 2 h in the dark at room temperature with the secondary antibody and were then washed thrice for 5 min in 1X PBS. Then they were rinsed in double distilled H2O followed by 100% ethanol and left to air dry. Coverslips were then mounted onto microscope slides using DAPI fluoromount and left for 1 h in the dark before imaging. Imaging was performed on an Axio Zeiss Imager.M2 microscope and on a Leica SP8 confocal microscope.

## MTT assay

Metabolic activity was analysed 48 h after cell seeding (Thiazolyl Blue Tetrazolium Bromide, Sigma-Aldrich, #M5655). Absorbance was measured at 560 nm using a microplate reader (Mithras LB 940, Berthold technologies). Values from the blank measurements were subtracted from the average based on six technical replicates. Five biological replicates were obtained.

## Strand-seq

Strand-seq libraries were generated as described previously[51] and outlined below. In brief, *TP53KO + SETD2KO* and wild-type neural stem cells were labelled with 40 μM BrdU for one single round of cell division. Cells were frozen and kept at −80 °C until further use. Cells were thawed in DMEM/F12 medium, centrifuged and resuspended in Nuclei Staining Buffer A (1.0 mL of 1 M Tris–HCl, pH 7.5, 308 μL of 5 M NaCl, 10 μL of 1 M CaCl$_2$, 5 μL of 1 M MgCl$_2$, 266.5 μL of 7.5% BSA solution, 100 μL of 10% (vol/vol) NP40 and 10 μL of 10 mg/mL Hoechst 33,258 to 8.3 mL of water) to a concentration of $1 \times 10^6$ cells/mL. Cells were filtered through a cell strainer and were kept on ice for ~30 min. Cells were then sorted in a 96-well plate with 5 μL ProFreeze-CDM freeze medium per well and stored at −80 °C. Following thawing, DNA was fragmented with 0.5 U Micrococcal Nuclease in MNase buffer supplemented with 1.5 mM DTT and 5% PEG 6000 for 8 min at room temperature in a final volume of 15 μL. The reaction was subsequently stopped with a final EDTA concentration of 10 mM. DNA was purified using AMPureXP beads at a 1.0x ratio and eluted in 10 μL EB Buffer on a BravoNGS system. Following elution, end-repair was performed for 30 min at room temperature using T4 DNA polymerase, Klenow and T4 PNK (all NEB) in T4 ligase buffer supplemented with dNTPs and the DNA was subsequently purified with AMPureXP Beads at a 1.8× ratio, using the BravoNGS system. After End-repair, DNA was A-Tailed for 30 min at 37 °C with Klenow exo- (NEB), followed by another round of bead purification with AMPureXp Beads at a 1.8× ratio. Forked Illumina adaptors at 33.5 nM final concentration per cell were ligated using quick ligase (NEB), followed by a AMPUre XP bead clean-up with 9.5 μL elution volume. DNA was incubated with 10 μg/mL Hoechst for 15 min and the plate was then irradiated with UV light at a total dose of $2.7 \times 10^3$ J/m$^2$ (using a crosslinker equipped with 5 × 365-nm longwave UV bulbs for 15 min). Oligonucleotides with a 6 bp multiplexing barcode (Sigma), specific to each well of a 96-well plate, were used with Primer PE 1.0 (Illumina) to amplify the nicked DNA with the Phusion HF

Master Mix (NEB) and the following programme: 98 °C for 30 s followed by 18 cycles of 98 °C for 10 s, 65 °C for 30 s and 72 °C for 30 s. Following amplification, all wells were pooled and purified as one pool with AMPureXP Beads at a 0.8× ratio to exclude free primers and adaptor dimers.

## Computational methods

**Whole genome sequencing and variant calling.** Whole-genome sequencing data and whole-exome sequencing data were processed by the DKFZ OTP pipeline[52]. Briefly, this workflow is based on BWA-MEM (v0.7.15) for alignment, biobambam (https://github.com/gt1/biobambam) for sorting and sambamba for duplication marking. Copy number variants were called using ACESeq[53] and structural variants were called by Delly(v1.1.6)[54] based on the aligned genomes. ACESeq output was used only for ShatterSeek[3] and for all other analyses copy number variants were called using CNMops(1.32.0)[55] with a 20 kb bin size, in combination with GC content correction and replication timing correction provided by ACESeq. DNAcopy algorithm was used for copy number segmentation.

## Inference of chromothripsis in bulk WGS data

Chromothripsis scoring of whole genome sequenced tumours were performed by ShatterSeek. Copy number variants from ACESeq[53] and structural variants from Delly[54] were provided as input to ShatterSeek. We applied the multivariable decision criteria from previous studies to define chromothripsis positive chromosomes[56] from ShatterSeek output.

## Inferring ecDNA fragments by AmpliconArchitect

To infer ecDNAs by AmpliconArchitect, we provided AmpliconArchitect[57] genome segments with copy number >=3 and the tumour alignment as input. AmpliconArchitect was allowed to explore other genomic regions connecting to the candidate genomic segment in an attempt to construct a circular amplicon. The output from AmpliconArchitect was filtered by removing circular segments with average copy number less than 3 and non-circular segments.

## Bulk RNA-seq gene expression analysis

Bulk whole transcriptome sequencing data were processed and normalised by kallisto[58]. GENCODE basic version 30[59] along with Human genome reference GRCh38 were provided to kallisto as reference. Kallisto reported expression levels per transcript. The expression across transcripts were summed to produce a gene-level expression measurement in transcripts per million (TPM) and in raw count. We used DESeq2 for differential gene expression analysis[60]. Genes were filtered if they did not exceed 10 counts in 3 or more samples to ensure good quality data. We investigated two distinct sample sets, one generated from 46 fresh frozen[39] tumour samples and the other from 173 FFPE[40,41] tumour samples. For the former, we compared chromothripsis positive medulloblastomas to non-chromothriptic medulloblastomas (with CT status determined from[61]), while in the latter, *TP53* mutation status was taken to mean CT+ (based upon a close to 100% CT rate in SHH *TP53* mut medulloblastoma). In both cohorts, the tight link between *TP53* mutation and CT made it impossible to control for *TP53* status when comparing CT+ to CT− tumours. To account for influences from normal cell types in the tumour cell population, we used the data from Riemondy et al. to create normal cell type specific gene signatures, using the top 100 significantly differentially expressed genes per cell type. These genes were removed from the expression matrices, subsequently performing a multivariate analysis adjusting for tumour cell content as well as SHH subgroup (Supplementary Data 6). Although the removed genes may well have an important role in medulloblastoma biology, we take this conservative approach to ensure that the derived signal is tumour-specific. Following this, the top 2000 up- and down-regulated genes were subsequently used for GSEA enrichment analyses, equivalent to the single-

cell data analysis described in *'Differential gene expression analysis and Gene Set Enrichment Analysis for individual copy number clones'* (see below).

## Copy number assessment of exome sequencing data

Control-FREEC version 11.4[61] was used for copy number assessment of the high coverage exome sequencing data. Known SNPs from dbsnp v142 were used as reference. The analysis was restricted to the exome capture region without the untranslated regions (UTRs).

## Phylogenetic inference from bulk sequencing data

Phylogenetic inference was based on SNVs and CNVs using Expectation-Maximisation on a multinomial model as previously published in Körber et al., 2019[62]. A few adjustments were made to the inference algorithm, in order to account for multiple samples. These adjustments are outlined in the following and an updated version of the code is available on github (https://github.com/hoefer-lab/phy_clo_dy/tree/master/multi_sample).

**Input data.** We used the read counts at all SNVs passing the quality filters for tree learning. If a mutation was absent in at least one sample, we manually checked whether the mutation was present in that sample but did not pass the filtering criteria and adjusted the input data accordingly. Coverage ratios were looked up at each mutated position using the output of Control-FREEC. In order to map copy number changes that did not carry an SNV to the tree, we additionally added all loci at which the coverage ratio changed in at least one sample to the input. If these positions did not harbour a mutation, we set the reference read counts to the average coverage across all mutated sites and the mutated read counts to zero. Moreover, we added the location of *TERT* and *PTEN*, as well as locations on chromosome 2p, 3p, 3q, 5p, 7p, 7q, 10p, 10q, 11p, 16q and 17p to map gains and losses on these chromosomal arms to the phylogenetic tree. The positions were taken as the midpoint of the gained or lost segment according to the output of Control-FREEC.

**Candidate trees.** The number of binary trees grows fast with the number of clades and thus finding the best solution in a multi-sample problem requires an efficient searching strategy. We here addressed this problem by initiating the algorithm with a set of candidate trees, which were based on prior information and sequentially expanded during the fitting procedure.

A first set of candidate trees was based on the mutational spectrum across the three samples. Mutations were either shared by all samples, or by the primary tumour and the metastasis only, or were private to a single sample. Thus, the mutation spectrum is consistent with a phylogenetic tree setting the relapsed tumour apart from the primary tumour and the metastasis. This is the simplest tree that is in agreement with the data from a combinatorial point of view. In order to account for more complex solutions, we extended this basal tree to more complex candidate trees by splitting individual clades.

Second, we took all unique trees consisting of up to five clades and split each clade into three subclones, corresponding to the three samples. These trees were extended by adding clades above each node during the optimisation algorithm and accepted if they yielded an improved solution based on a Bayesian Information Criterion. Extensions were abrogated if each sample consisted of three subclones or if the solution did not improve.

**Additional adjustments.** As compared to the algorithm described in Körber et al., 2019[62], we added a few additional adjustments:

Model selection was based on a modified Bayesian information criterion as outlined in Körber et al., 2019, but without prior tree selection based on clonal mutation estimates.

We accounted for the possibility that a mutation call was false negative in the candidate tree (i.e., truly present, but not detected in a sample).

We restricted the range of normal copy numbers from [0.9, 1.1] to [0.95, 1.05].

We added prior information on whether a copy number change observed in multiple samples was likely due to a single event based on manual inspection of the copy number profiles. Specifically, we required that the losses on 2p, 3p, 11p, 16q and 17p, as well as the gains on 3q were due to single events.

## DNA methylation

The majority of the DNA methylation profiles were published in a previous study[63]. Genomic DNA was extracted from fresh-frozen or formalin-fixed and paraffin-embedded (FFPE) tissue samples. DNA methylation profiling of all samples was performed using the Infinium MethylationEPIC (850k) BeadChip (Illumina, San Diego, CA, USA) or Infinium HumanMethylation450 (450k) BeadChip array (Illumina). All computational analyses were performed in R version 3.5.3 (R Development Core Team, 2021; https://www.R-project.org). Raw signal intensities were obtained from IDAT-files using the minfi Bioconductor package version 1.21.4[64]. Illumina EPIC samples and 450k samples were merged to a combined data set by selecting the intersection of probes present on both arrays (combineArrays function, minfi). Raw methylation signals were normalised by the function preprocessIllumina. Possible Batch-effects caused by the type of material tissue (FFPE/frozen) and array type (450k/EPIC) were adjusted by fitting univariable, linear models to the log2-transformed intensity values (removeBatchEffect function, limma package version 3.30.11). The methylated and unmethylated signals were corrected individually. Beta-values were calculated from the back-transformed intensities using an offset of 100 (as recommended by Illumina). Filtering of CpG probes was performed as described in Capper et al. 2018[63]. In total, 428,230 probes were kept for downstream analysis. To perform unsupervised non-linear dimension reduction, PCA was applied to the 50,000 probes with highest standard deviation and the resulting first 100 PCs were used for UMAP analysis (R package uwot 0.1.8). The following non-default parameters were applied: n_neighbors = 10; min_dist = 0.5.

## scDNA-seq data pre-processing and quality control

The raw base call files from the 10X Chromium sequencer were processed utilising the Cell Ranger DNA (version 1.1.0) pipeline for alignment and cell calling. First, the "cellranger mkfastq" command was used to demultiplex the sequencing samples and to convert barcode and read data to FASTQ files. Then, the "cellranger-dna cnv" command was used to perform reference alignment and cell calling. As a reference genome we used pre-build Human reference GRCh37 (hg19), which was downloaded from 10X genomics website (version 1.0.0 from June 29 2018, https://support.10xgenomics.com/single-cell-dna/software/downloads/latest).

Default cell calling parameters as implemented by Cell Ranger DNA were used.

## Copy number inference in single nuclei/cells

We inferred single cell Total Copy Number and median cell ploidy using the scAbsolute pipeline[27] based on the bam files created by the cellranger DNA pipeline (above) for subsequent identification of genetic clones within each sample (see below). Initially, we utilised cellranger-dna bamslice to create bam files for every individual cell in each sample. Only non-duplicated primary and well aligned reads were kept according to the following flags: read unmapped (0 × 4), not primary alignment (0 × 100), read fails platform/vendor quality checks (0 × 200), read is PCR or optical duplicate (0 × 400), as well as supplementary alignment (0 × 800) and a mapping quality ≤ 30. The filtered BAM files were used as input for the scAbsolute workflow, as described

by Schneider et al.[27], at a bin size of 500 kb for all samples except LFS-MBP Nuclei, where a bin size of 1 Mb was observed to provide better GC and mappability correction. The pipeline output includes ploidy estimates per cell, segmentation of the read counts per genome for each cell, and resulting copy number estimates per segment in each cell.

Following the recommendation of the scAbsolute authors[27], the resulting segmentation and copy number estimates were manually reviewed for each cell to validate the correct fit of the scAbsolute model. Manual review identified several cells within the LFS-MBP Nuclei sample for which regions in the genome had read-counts falling in-between integer copy number states, suggestive of incorrect ploidy estimates (Supplementary Fig. 1a). These cells had an initial median ploidy estimate of 2, and an average copy number of less than 2. Alternative fitting without allowing the ploidy 2 solution resulted in an assignment to a ploidy of 4. Manual review found these regions to be segmented and modelled with qualitatively better fit to integer copy numbers at the higher ploidy (Supplementary Fig. 1a). As that the majority of cells from the LFS-MBP Nuclei sample were estimated to have a ploidy of 4, we accepted this as the more parsimonious solution. Finally, we removed any cells which were outliers (greater than 2x MAD from the median value) based on overdispersion of read counts across bins (evaluated as the ratio of observed variance across bins compared to the expected variance under a poisson count model) or normalised gini coefficient (Supplementary Fig. 1b–d).

## Clonal inference from single-cell DNA-seq data

To identify copy-number clones within the single-cell data, we first mapped the copy number states of each cell/nucleus from the sample onto a set of common segments. The common segments were defined by taking all unique breakpoints estimated by the scAbsolute pipeline across the cells within each sample, defining chains of breakpoints within 2 bins of another breakpoint, and collapsing such chains to a single breakpoint at the median location within the chain. The copy number state for each cell and segment was then taken to be the median copy number state across the segment, resolving ties by taking the value closer to the cell ploidy (as estimated by scAbsolute).

We then split each sample into groups of cells with the same ploidy and performed Ward D2-linkage hierarchical clustering within these ploidy groups, with Manhattan distance on the common segments. We cut the resulting trees into k clusters, evaluating k from 1 to 12, and only keeping clusters with 5 or more cells. Cells which belonged to a ploidy group with less than 5 members were excluded from the subsequent analysis. For each cluster, we estimated pseudobulk copy number profiles as described below. Following this, we bootstrapped the cells assigned to the cluster, inferring 101 bootstrapped CNV profiles per cluster. We then chose the number of clusters k by examining two metrics: First, for the observed cell assignment we calculated a goodness of fit for the resulting pseudobulk Total Copy Number profile, reasoning that clusters with heterogeneous populations will be poorly modelled by integral copy number states, and computed the BIC, taking into account the number of estimated copy number states across each cluster for each value of k. We also performed an F-Test on the distributions of bootstrapped CNV profiles, testing whether the within-cluster distances (calculated with Euclidean distance) were different from the between cluster differences, and examined the resulting p-values for deviation from the null. These two complementary statistics provide measures of uniformity within clusters and heterogeneity across clusters respectively. The local-minima of these two statistics provided candidate values for k, which we evaluated by manual review of both the resulting copy number fits to the pseudobulk read counts for each cluster, as well as through examining the assignment of cells to clusters in dimensionality reduced embeddings generated using tSNE[65] and multidimensional scaling[66]. When multiple numbers of clusters looked similarly plausible, the smaller and more conservative k was chosen to prevent overclustering. The selected k values for all samples

across all ploidies are shown in Supplementary Data 1, and the resulting number of cells per cluster is shown in Supplementary Data 3.

## Copy number inference for pseudobulk of clusters

Independently from the procedure described for inference of cell-specific copy number using scAbsolute, for each cell in each sample, we followed the instructions of HMMCopy-utils to count the number of deduplicated reads aligning to each 20 kb bin across the genome. For each cluster and assignment of cells to the cluster (including bootstrapped samples as described above), we estimated a pseudo-bulk profile by summing the read counts falling within each bin across all cells assigned to the cluster, and corrected the read counts to obtain logR ratios using a loess model for GC and mapability bias as implemented by HMMCopy[67]. Each pseudobulk profile was segmented and copy number was estimated using the model implemented in HMMCopy. The ploidy estimate for the cluster from scAbsolute, as well as the observed copy number states from the coarse-grained scAbsolute segmentation, were used to set the number of available copy number states, as well as the prior on the μ parameter for each state. To encourage longer segments and to prevent over-segmentation, which may lead to false chromothripsis calls, we used prior values of e = 0.999999999999999, and strength = 1e + 50.

## Chromothripsis (CT) detection at the single clone level

We adjusted established criteria for inferring chromothripsis in cancer genomes from bulk whole-genome sequencing[3,4]. Our pseudobulk CNV data consists of copy state per bin (20 kb) across the genome, divided into chromosomes. We first removed any copy number state that was present for only a single consecutive bin. To detect CT in a clone, we then looked for 50 Mb windows with 8 or more copy number switches. We assessed every 50 Mb window across the 20 kb bins with a sliding window approach. As a chromosome-level CT score, we calculated the fraction of evaluated windows in a given chromosome that are determined as CT positive. The sliding window calculation was sped up using Fourier Transform implementations of convolution operators, as implemented by the R programming language[68].

In addition to a chromosome level CT score, we computed a bin level score to estimate the boundaries of CT regions. We calculate the bin CT score as the fraction of windows containing that bin which are called CT positive as described above. This score is assigned to every bin but represents the level of CT in a 50 Mb neighbourhood, centred in the bin of interest. We note that there is a border effect due to bins in the limits of the chromosome being evaluated less times and hence the numerical CT score may be less precise in these regions. The bins which had a positive bin-level CT score were considered to define the regions of CT in the samples.

In addition to CT scoring the HMMcopy inferred pseudobulk CNV profiles for each clone, we also scored the 101 bootstrapped CNV profiles inferred for each clone as described above. For downstream analysis, we considered CT events as high-confidence only if one of two criteria was true:

1. A CT event was detected on this chromosome for every clone in the sample (and therefore was a clonal CT event); regardless of how often the CT events in any of the clones were reproduced in the bootstrap samples;
2. If the event was not observed in all clones, at 51 of the 101 bootstrap samples (>50%) from the respective clone had a non-negative CT score for this chromosome.

Criteria 1 was specifically chosen to favour calling events as clonally CT even if subclones consisting of few cells had low confidence in detection of these events, to avoid over-detection of instances of subclonal CT.

The bootstrap filter was applied only on the level of calling particular chromosomes as CT; for all analyses where regions of CT were

compared with CNV segments, gene locations, or other genomic regions, the observed sample (and not the bootstrapped samples) was used to score each individual bin as CT positive as described above.

## Subsampling to estimate the sensitivity of the chromothripsis scoring

To evaluate the sensitivity of our CT scoring strategy, we subsampled the number of cells assigned to each clone across the MB243, LFS-MBP Nuclei, LFS-MBP PDX and LFS-MB1R PDX samples (those samples with subclonal CT detected), taking 101 random samples subsamples at 3, 4, 5, 6, 10, 15 and 20 cells per clone, and calling CT using the strategy described above.

We took every clone-chromosome combination with a high-confidence chromothripsis event prior to downsampling as the reference positive set. For each positive event in the reference set, we calculated the sensitivity as the number of bootstrap samples where the same chromosome-clone combination was scored positive. We then computed the median and interquartile range across the positive event set at each number of subsampled cells as an estimate of the sensitivity across the different events observed in our data.

## Assessment of chromothripsis scoring specificity

To assess the specificity of our CT scoring approach, we used the 10x-Chromium single cell CNVKit sequencing dataset of RPE-1 cells with CRISPR-induced genomic instability from[16], where targeted genomic stability was induced through telomeric loss specifically on chromosome 4. Downloading the aligned BAM file as processed by the original study authors from the SRA Run SRR10947879, we estimated clonal substructure and estimated clone-specific CNV profiles and conducted CT scores following the same steps as described above (scAbsolute to infer cell-level CNVs; clustering and bootstrapping to infer subclones; and ploidy-informed HMMcopy to infer clonal copy-number profiles). While the experimental setup does not guarantee the induction of chromothripsis in all cells within this sample, or completely exclude the possibility of genomic instability on other chromosomes, the approach employed provides a close real-world approximation of a dataset with known ground truth (due to the targeted locus and available matched WGS) for evaluating the specificity of the 10x Chromium scDNA-seq assay combined with our CT scoring approach.

## scRNA-seq data pre-processing and quality control

The raw base call files from the 10X Chromium sequencer were processed using the Cell Ranger Single-Cell Software Suite

(release v3.0, https://support.10xgenomics.com/single-cell-gene-expression). First, the "cellranger mkfastq" command was used to demultiplex the sequencing samples and to convert barcode and read data to fastq files. Based on the fastq files, "cellranger count" was executed to perform alignment, filtering, as well as barcode and unique molecular identifier (UMI) counting. The reads from single-nuclei RNA-sequencing were aligned to the pre-mRNA hg19 reference genome, while the reads from single-cell RNA-sequencing were aligned to the hg19 reference genome, implementing a pre-built annotation package downloaded from the 10X Genomics website. For all single-cell RNA-sequencing data resulting from PDX samples, we also mapped the reads to the mouse genome (mm10) in order to check whether cells map better to human or mouse. If less than 1% of the reads were aligned to hg19, we defined the respective cells as mouse cells. The filtered genes x cells matrix was further used as input for the data processing workflow described in the following.

## Analysis of single sample scRNA-seq data using scanpy

The output from the Cell Ranger was analysed with the scanpy software toolkit in python[69]. First, genes that were expressed (>=1 count) in <=5

cells across the whole dataset were removed (sc.pp.filter_genes with min_cells = 5). Next, we filtered single-cells and single-nuclei data individually. For single-nuclei, we filtered them for (i) counts (500 <total_counts <25,000), (ii) genes (300 <n_genes <6000), (iii) mitochondrial genes (pct_counts_mt <5%) and ribosomal genes (pct_counts_ribo <10%). Single-cells were filtered for (i) counts (500 <total_counts <25,000), (ii) genes (200 <n_genes), (iii) mitochondrial genes (pct_counts_mt <10%) and ribosomal genes (pct_counts_ribo <40%). In addition, we used scrublet[70] to remove potential doublets in our dataset, see Supplementary Data 4 for details). To account for variable sequencing depth across cells, we normalised unique molecular identifier (UMI) counts by the total number of counts per cell, scaled to counts per 10,000 (CP10K; sc.pp.normalise_per_cell), and log-transformed the CP10K expression matrix (ln[CP10K + 1]; sc.pp.log1p). Next and to generate cell type clusters, we selected the 2000 most variable genes across samples by (1) calculating the most variable genes per sample and (2) selecting the 2000 genes that occurred most often across samples (sc.pp.highly_variable_genes). After mean centreing and scaling the ln[CP10K + 1] expression matrix to unit variance, principal component analysis (PCA; sc.tl.pca) was performed using the 2000 most variable genes. To select the number of PCs for subsequent analyses, we used a scree plot and estimated the "knee/elbow" derived from the variance explained by each PC. Visualising the data in a UMAP embedding showed good alignment across normal cell types, while tumour cell populations clustered separately. Hence, we did not perform correction for sample specific batch effects, following the recommendations of Luecken et al.[71]. Following this, we calculated clusters using the Leiden graph-based clustering algorithm v0.7.0[72], which were subsequently used for differential gene expression as described in the following. Clustering stability was post-hoc validated using bootstrap resampling and SCCAF[70].

### Differential gene expression analysis and cell cluster annotation

To evaluate the cellular identity of distinct clusters, we annotated them based on the expression of known cell marker genes collected from the literature[35–38]. For this purpose, we performed a two-sided Wilcoxon rank-sum (Mann-Whitney-U, Benjamini-Hochberg adjusted) test to compare each individual cluster to all other cells. Next, we then used the mentioned list of genes to assign cell identities to specific clusters. This list of known marker genes included *CD74, SAT1, MERTK* (macrophages/microglia), *VWF, EGFL7, INSR* (endothelial cells), *COL4A1, FN1, CDH11* (meninge cells), *FABP7, CLU, GFAP, SLC1A3, PTN* (astrocytes), *CD74, HLA-DRB1, RGS1, LYZ, CD81* (microglia), *GLI2, PTCH2, HHIP, POU6F2* (malignant SHH), *MKI67, TOP2A, DIAPH3, POLQ* (malignant cycling), *PTPRD, MARCH1, NCAM2, PLCL1, NEUROD1* (malignant neuronal development I) and *RASGEF1B, SLC26A3, LINGO1* (malignant neuronal development II). In cases where the identity could not be resolved, the highest variable genes were used as input for a CellMarker database search (http://biocc.hrbmu.edu.cn/CellMarker/[73]). Alternatively, the ToppGene suite (https://toppgene.cchmc.org/) was used to evaluate the cellular identity of a cluster[74].

### Cell of origin analysis in single-cell RNA-sequencing data

To investigate the likely cell of origin for our tumour samples, we compared the expression of tumour cells to an atlas of normal cell types and states. For this purpose, we accessed the publicly available dataset from Aldinger et al., comprising 21 distinct normal cell types and states, and performed a pairwise correlation analysis of the transcriptome of each annotated cell state with the cell types from Aldinger et al.[38] (Supplementary Fig. 3.2). For each malignant cell state, the Pearson correlation of the intersection between the genes expressed in the malignant cells and significantly differentially expressed genes ($p$-value ≤ 1e$^{-5}$) for the respective Aldinger cell type was calculated.

### Projection of scRNA-seq data onto non-LFS medulloblastoma samples

To assess the extent of transcriptional resemblance between LFS and non-LFS SHH MBs, we leveraged publicly available data from Vladoiu et al.[36] and projected our samples into a shared UMAP embedding using ingest following the scanpy vignette (https://scanpy-tutorials.readthedocs.io/en/latest/integrating-data-using-ingest.html).

### Single-cell RNA-seq copy number detection by inferCNV

We inferred copy number variation in single-cell RNA-seq data using inferCNV[42]. The quality-controlled count matrices (see Analysis of single sample scRNA-seq data using scanpy above), separately for each single sample, were used as input. For the patient-derived single-nuclei samples, we used endothelial cells from the same donor as a reference cell population, as we had these data for each donor. We used these to define the diploid reference level. For the PDX samples, we used the closest available reference to a diploid state. To avoid outsized impact of any particular gene on the copy number estimate, the gene expression values were limited to [−3,3]. We then estimated the underlying probability of each CNV using the HMM and Bayesian Network methods implemented by inferCNV. In addition to the dynamic de-noising implemented in inferCNV, an additional median filter was used when visualising the modified expression values output by the method.

### Integrating scDNA- and scRNA-seq data based on copy number profiles

To project our scRNA-seq data on the clones defined using the scDNA-data, we first discretized the modified gene expression values as calculated by inferCNV into 3 states: 1 copy, for genes where modified_expression <1.0 − sd(modified_expression); diploid, for genes with 1.0 − sd(modified_expression) ≤ modified_expression ≤ 1.0 + sd(modified_expression); and 3 copies for genes where modified_expression >1.0 + sd(modified_expression). We then mapped each clonal scDNA CNV profile to gene-level copy number. For clones with a median ploidy not equal to 2, we divided the CNV profile to achieve a median 2 copy state, as whole genome doubled populations were not effectively detected in our scRNAseq data.

As many chromosomes did not harbour subclonal alterations in our samples, we filtered the scRNA and scDNA derived CNV data to variable chromosomes. To do so, we counted the number of distinct copy number values per chromosome arm and clone, resulting in a table of clones x copy number values. Using this table, we calculated the frequency of CNVs per clone per chromosome, which was subsequently utilised to assess the standard deviation in the CNV distribution across clones. If a chromosome of interest showed a variability greater than 15%, we included it in our correlation approach, while all other chromosomes were discarded.

We further visually examined the variation across each chromosome in scRNA-seq. If a chromosome showed no variation in the scRNAseq, i.e. did not contain CNVs or did not display heterogeneous CNVs, the respective chromosome was excluded from the downstream correlation.

Using the subset genes mapped to these chromosomes, we calculated the Pearson correlation between each copy number profile from the scDNA-seq clones and each cell in the scRNA-seq data ($R_o^2$). We assessed statistical significance using a permutation testing approach, creating randomised CNV profiles by permuting across clones the copy state for each gene independently in the scDNA data (10,000 permutations). Each cell was then assigned to a scDNA clone according to the maximum correlation if the Bonferroni corrected $p$-value (corrected for number of clones in sample) was smaller than 0.05.

We used the difference in correlation between the best and 2nd best matching clone correlation as a measure of uncertainty for each of the cell-to-clone assignments. If the difference was ≥0.025, we determined a cell as being confidently assigned to a clone; values below were

considered insecure assignments. If 50% of cells assigned to a clone could not be confidently assigned and more than 50% of the insecure cells can be confidentially assigned to another clone, cells from both clones in scRNA-seq were merged into one group for downstream transcriptional analysis. This merging was overturned for clones where clear copy number differences could be detected in visual inspection of the inferCNV modified expression values (e.g. chr22 loss in Clone5 for LFS-MBP PDX). The distance between best and 2nd best matching clone is visualised as uncertainty in Supplementary Figs. 13 and 14.

### Differential gene expression analysis and gene set enrichment analysis for individual copy number clones

Using the copy number clone information generated as described above, we performed differential gene expression analysis between each merged copy number clone and all other cells. The two-sided Wilcoxon rank-sum (Mann-Whitney-U, Benjamini-Hochberg adjusted) test from scanpy was used in order to identify significantly up-regulated genes. For individual copy number clones with at least one significantly differentially expressed gene, the complete gene list was combined with the logFC values and used as input for gene set enrichment analysis (GSEA) as implemented in the R package HTSanalyzeR2 (https://github.com/CityUHK-CompBio/HTSanalyzeR2,[75]). Thereby, GSEA was performed using the MSigDB hallmark gene sets provided by Liberzon and colleagues[76]. The results from GSEA were filtered according to the underlying $p$-value (FDR < 0.05, Kolmogorov–Smirnov statistic, Benjamini Hochberg adjusted). Hence, if a pathway was significantly altered ($p$-value < 0.05), it was kept in the analysis, while non-significant pathways were discarded ($p$-value > 0.05). Subsequently, the clones and altered pathways were visualised using a custom R script.

### Evaluating druggable targets from scDNA- and scRNA-sequencing data

We assessed differences in clonal expression of druggable targets using the projection of scRNAseq data onto clones described above. To increase sensitivity for between-clone differences, we further filtered any cells where the correlation difference between the best and second best matching clone was less than 0.025. In cases where the second best matching and best matching clone were merged, the distance to the third best clone were used.

CNVs within bins corresponding to genes reported as potential druggable targets by Worst et al. 2016 et al.[43] were evaluated across clones to evaluate the presence or absence of focal gains. For per-cell visualisation in Fig. 3i, we used the log ratio of the observed read counts/ median read counts in the autosomes, averaged across all bins falling overlapping each gene. For Supplementary Figs. 13 and 14, we used the clonal CNV state as inferred by the procedure described above, taking the average copy number across the two clones when two clones were merged according to the criteria in the previous section.

We investigated the expression of druggable targets from scDNA-seq (see above) in the scRNA-seq data to evaluate whether we can observe transcriptional consequences. Hence, we used the normalised expression matrix from scRNA-seq for each sample and subset it to the druggable target genes. Then, we used the integrated clone as well as the cell type information for normal cells to compare the expression of each of these groups of cells. Importantly, we removed cells with no expression in a respective gene as well as genes which were not expressed in at least 50 cells.

### R2 genomics analysis visualisation platform

R2 Genomics Analysis Visualisation Platform (Website: https://r2platform.com) was used to compare survival data and generate Kaplan-Meier Plot using the data set Tumour Medulloblastoma – Cavalli – 763 – rma_sketch – hugene11t, a minimal group size of 10, and separating by a single gene (*SETD2*). More specifically, shh_alpha was selected for the subtype.

### Statistics and reproducibility

Specific statistical tests used in this study are described in detail in the methods and indicated in the respective figure legends or in the main text, where appropriate. $P$ values of less than 0.05 were considered as significant. Multiple testing corrections were applied whenever mentioned. No statistical method was used to predetermine sample sizes. Statistical comparison of the effect of *SETD2* and *TP53* knockouts in neural stem cells (Fig. 6b, c, e, h) was based on a minimum of three biological replicates. For experiments with varying numbers of biological replicates/group (Fig. 6e, h), no data were excluded from the analyses. Variable sample size was due to different growth rates of cells, limiting their availability at the time of the experiments.

### Reporting summary

Further information on research design is available in the Nature Portfolio Reporting Summary linked to this article.

## Data availability

Raw sequencing data from single cell WGS, single cell RNAseq, and Strand-Seq experiments generated in this study are deposited with the European Genome-phenome Archive (EGA), which is hosted by the EBI and the CRG, under accession number EGAS00001005410. The data are available under restricted access due to the European General Data Protection Regulation (GDPR) and the German General Data Protection Regulation (GDPR) and to respect the patient consent forms. Data access can be requested through the EGA subject to Data Access Committee review. It can be granted in principle for research use after a Data Transfer Agreement is legally settled between the requesting institute and the providing institute. Once the data access has been granted, the access is usually available for 5 years, unless otherwise restricted by individual patient consent forms. Data access requests will be reviewed and Data Transfer Agreements will be settled as quickly as possible.

Processed sequencing data for WGS/WES, scDNA, scRNA and StrandSeq assays are available on Zenodo under the https://doi.org/10.5281/zenodo.13348419. The samples profiled in this study are embedded in the larger ICGC PedBrain project, and raw sequencing data for all analyses of bulk short read WGS and RNA sequencing for the PedBrain samples are available after through the EGA under accession number EGAS00001001953[39]. Raw RNAseq data used for differential expression analysis in this study from Waszal et al. is available through the EGA under accession number EGAS00001004126[40] and from Kool et al. from the EGA under accession number EGAS00001000607[41]. scRNAseq data from Reimondy et al. use in this manuscript are available from GEO under SuperSeries accession number GSE156053[23]. Raw data for scRNAseq from Vladoiu et al. used in this manuscript are available from the EGA under accession code EGAS00001003170[36]. Raw snRNAseq data from Aldinger et al. used in this manuscript are available from dbGaP under accession code phs001908.v2.p1[38]. Raw single cell WGS data from Umbreit et al. used in this manuscript are available from the Sequence Read Archive under project code SRP243832[16].

The methylation array data from Capper et al. used in this manuscript are available through GEO under accession number GSE109381[63].

Source data for all figures are accessible through Zenodo under the https://doi.org/10.5281/zenodo.13918598. The remaining data are available within the Article, Supplementary Information or in the Source Data files.

## Code availability

The aforementioned computational methods provide a summary of the procedures implemented in various custom-made R, python and bash scripts. These scripts contain the commands run for the analyses highlighted in this publication. In order to sustain reproducibility, they are publicly available on Github (https://github.com/PMBio/MB_scSeq).

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

## Acknowledgements

We thank Peter Lichter and Aurelio Teleman for discussions, Frauke Devens, Michaela Hergt, Brigitte Schoell and Katharina Bauer for technical support, the Sequencing and the Microarray units of the Genomics and Proteomics Core Facility (DKFZ), the EMBL Sequencing Facility, the DKFZ FACS Core Facility and the DKFZ Imaging Facility. David Pellman and his team are acknowledged for kindly sharing their published single-cell DNA sequencing data as a reference dataset. Florian Markowetz and his team are acknowledged for sharing scAbsolute and for support in using it. Thomas Weber and Jan Korbel are acknowledged for their advice regarding strand-seq data analysis. Axel Benner is acknowledged for support with statistical analyses. Daniel Haag is acknowledged for kindly sharing neural stem cells. P.S. was supported by the Heidelberg-Mannheim Life Science Alliance. D.R.G. was supported with personal grants by the German Academic Scholarship Foundation (Studienstiftung des Deutschen Volkes) and the Mildred Scheel Doctoral Fellowship programme of the German Cancer Aid (Deutsche Krebshilfe). K.W.P. acknowledges funding by the German Childhood Cancer Foundation (DKS2021.02) and the Federal Ministry of Education and Research (01GM2205A). R.G.P. holds a fellowship from Grant IHMC22/00007 funded by the Instituto de Salud Carlos III (ISCIII). The vast majority of the experimental work in this study was supported by grants to A.E. from the DFG, the Wilhelm Sander Foundation and the Fritz Thyssen Foundation. H.S. was supported by the German Federal Ministry of Education and Research (031L069A).

## Author contributions

P.S., M.J.P., R.G.P. and H.S. performed the vast majority of the computational work, M.S.-L. performed the vast majority of the experiments, M.R. performed the initial single-cell experiments, J.K.L.W. contributed to the bulk sequencing data analysis, V.K. derived phylogenies from bulk sequencing data, P.M. contributed to the single-cell experiments, G.P. contributed to the strand-seq experiments, M.S. contributed to the DNA methylation analysis, T.K. supported the CRISPR/Cas9 experiments, R.K. did the confocal imaging, N.C. contributed to the computational work, K.O. contributed to the bulk RNA sequencing data analysis. D.R.G., K.K.M. and K.W.P. supported the nuclei extraction, A.J. did the image analysis of multicolour FISH, A.K. did the neuropathology evaluation of tissue, T.H. supervised the computational analyses of bulk phylogenies, M.Z. supervised part of the bulk sequencing data analysis, S.M.P. contributed by providing feed-back on the work and critical revision of the article, W.H. supervised the computational work and provided critical revision of the manuscript, O.S. had oversight, leadership responsibility and conceptualisation of the bioinformatic analyses, A.E. conceived the study and had leadership responsibility for the experimental work.

## Funding

## Competing interests

The authors declare no competing interests.

## Additional information

[1]Group Genome Instability in Tumors, German Cancer Research Center (DKFZ) and German Cancer Consortium (DKTK), Heidelberg, Germany. [2]European Molecular Biology Laboratory, Genome Biology Unit, Heidelberg, Germany. [3]Division of Computational Genomics and Systems Genetics, German Cancer Research Center (DKFZ), Heidelberg, Germany. [4]Wellcome Sanger Institute, Wellcome Trust Genome Campus, Cambridge, UK. [5]Life Sciences Department, Barcelona Supercomputing Center, Barcelona, Spain. [6]Division of Molecular Genetics, German Cancer Research Center (DKFZ), Heidelberg, Germany. [7]Division of Theoretical Systems Biology, German Cancer Research Center (DKFZ), Heidelberg, Germany. [8]Single-cell Open Lab, German Cancer Research Center (DKFZ) and Bioquant, Heidelberg, Germany. [9]Faculty of Biosciences, Heidelberg University, Heidelberg, Germany. [10]Hopp Children's Cancer Center Heidelberg (KiTZ), Heidelberg, Germany. [11]Division of Pediatric Neuro-oncology, German Cancer Consortium (DKTK) and German Cancer Research Center (DKFZ), Heidelberg, Germany. [12]Department of Pediatric Oncology, Hematology and Immunology, Heidelberg University Hospital, Heidelberg, Germany. [13]Institute of Human Genetics, Heidelberg University, Heidelberg, Germany. [14]Clinical Cooperation Unit Neuropathology, DKFZ, Department of Neuropathology, Heidelberg University Hospital, Heidelberg, Germany. [15]These authors contributed equally: Petr Smirnov, Moritz J. Przybilla, Milena Simovic-Lorenz, R. Gonzalo Parra, Hana Susak. [16]These authors jointly supervised this work: Oliver Stegle, Aurélie Ernst. ✉e-mail: o.stegle@dkfz-heidelberg.de; a.ernst@dkfz-heidelberg.de

