## [Transparent Peer Review file · Nature Communications]

Multi-omic and single-cell profiling of chromothriptic medulloblastoma reveals genomic and transcriptomic consequences of genome instability

Corresponding Author: Dr Aurelie Ernst

This manuscript has been previously reviewed at another journal. This document only contains reviewer comments, rebuttal and decision letters for versions considered at Nature Communications.

Version 0:

Reviewer comments:

Reviewer #1

(Remarks to the Author)

Ernst and colleagues have undertaken a strong effort to reframe the manuscript, softening some conclusions, including alternative sequencing approaches and comparisons to published data to validate their scSeq conclusions, and they have also used a new dataset to analyze chromothriptic vs non-chromothriptic tumors. I enjoyed the manuscript, but request the following updates:

Lines 216-219: The claim is made that there is no correlation between the number of copy-number SEGMENTS and ecDNA copy number, referencing Fig 2d. However, Fig 2d references copy number STATES (ie the presence of segments of 1 copy, 2 copies, 6 copies being 3 copy number states). Which did you mean? It is also worth mentioning that subclonal chromothripsis was observed at this locus (Fig 1c), and clarifying how you know the chromothriptic chromosome was absent, while the ecDNA is present. In the last version of this manuscript, this was done by CT score.

Relatedly, in the Fig. 2d legend, you refer to “cnv states per cell as a proxy”. Did you mean CNV states on chr2, where the CT event is? And do you mean to say as a proxy “for defining CT at this locus”?

Lines 306-320: The source of these fresh-frozen and FFPE samples should be included, as well as how CT status was determined in the fresh-frozen samples. In the methods (lines 809-10) it is stated that for the FFPE samples “TP53 mutation status was used as an approximation of chromothripsis status”. This should be reworded to state that TP53 mutation was taken to mean CT+, based upon a 100% CT rate in SHH TP53 mut medulloblastoma.

Supplementary Table 3.3: What are the two tabs? Are they meant to be the fresh-frozen and FFPE analyses? If so, this should be stated. Also, I noticed that SHH is missing from the data despite its mention (line 313), and that RRP1B has a $0 < \log_2FC < 1$, which should not be included based upon the criteria listed in lines 310-11. It should also be explicitly mentioned in the text that “it cannot be excluded that these differentially expressed genes are not due to TP53 status and not chromothripsis”, assuming TP53 status is how you determined CT status.

Although compelling, the link between SETD2 and CT does not determine whether SETD2 causes chromothripsis or, whether like TP53mut, it allows CT cells to survive. To clarify this, lines 458-60 should be adjusted to say “...suggest a possible causative or permissive role for these two genes in the occurrence of CT in medulloblastoma.” To further emphasize this, it would also change “linked” in line 526 with “correlated”.

It is worth citing Espejo Valle-Inclan et al., Biorxiv 2023, as they have also shown detailed reports of subclonal chromothripsis in patient data, albeit not from single-cell data.

Reviewer #5

(Remarks to the Author)

Overall, I think the authors have done a reasonably good job addressing the concerns and criticisms raised by the previous

reviewers. Importantly, the authors have used an orthogonal scDNA-seq method (Picoplex) and conducted extensive benchmarks and experiments (e.g., FISH, time-course LFS cell cultures) to verify the robustness of their methods and conclusions. Regarding the limited sample size, considering the rarity of LFS medulloblastoma, a significant expansion of the number of patients is impractical, and the current datasets seem sufficient for drawing the current conclusions in this paper. I have two suggestions to further improve the manuscript.

1. Regarding the title, "single-cell multi-omics" is misleading as the scDNA-seq and scRNA-seq were performed for different cells (although for the same sample). Single-cell multi-omics usually refers to simultaneous omic profiling for the same cell. This study actually also included plenty of bulk sequencing datasets, so I suggest to remove "single cell" from the title.

2. Regarding the authors' response to Main Concern #10 from Reviewer #2 (integration of bulk RNA-seq and scRNA-seq data), I think the impact of microenvironments on the differential expressions is not fully resolved. I suggest the authors to use Scissor (Sun et al. Nat Biotechnol 2021) which combines bulk RNA-seq to identify the phenotype-associated cell types in scRNA-seq data.

Reviewer #6

(Remarks to the Author)

I carefully read your paper and rebuttal letter and I wish to acknowledge that you have addressed key concerns raised by the reviewers about this research in the revised version of your manuscript.

Version 1:

Reviewer comments:

Reviewer #1

(Remarks to the Author)

I have reviewed the manuscript and the authors have appropriately addressed my concerns. One comment that does not require re-review: In Fig. 2c and Supplementary Fig. 3.6, percent should be written out of 100, not out of 1.

Reviewer #5

(Remarks to the Author)

My concerns have been well addressed.

REVIEWER COMMENTS – July 2024

Reviewer #1 (Remarks to the Author):

Ernst and colleagues have undertaken a strong effort to reframe the manuscript, softening some conclusions, including alternative sequencing approaches and comparisons to published data to validate their scSeq conclusions, and they have also used a new dataset to analyze chromothriptic vs non-chromothriptic tumors. I enjoyed the manuscript, but request the following updates:

Lines 216-219: The claim is made that there is no correlation between the number of copy-number SEGMENTS and ecDNA copy number, referencing Fig 2d. However, Fig 2d references copy number STATES (ie the presence of segments of 1 copy, 2 copies, 6 copies being 3 copy number states). Which did you mean? It is also worth mentioning that subclonal chromothripsis was observed at this locus (Fig 1c), and clarifying how you know the chromothriptic chromosome was absent, while the ecDNA is present. In the last version of this manuscript, this was done by CT score.

We thank the reviewer for pointing out this inconsistency. The intended word here was indeed segments. We have now clarified that we use the number of CNV segments (which reflects the number of breakpoints) as a proxy for chromothripsis scoring, as we found that this is more robust as compared to CT scoring directly on single cells (please kindly refer to Supplementary Figure 1.8). Briefly, this is due to the fact that the accuracy of CT scoring on single cells depends on the specifics of the CT event in question (size, number of breakpoints, etc), as well as the sequencing depth of each individual cell. Nevertheless, we believe that the lack of breakpoints detected across the chromosome, together with the evidence that ecDNAs are independent of genomic clones, is indicative of cells that carry the ecDNA but no copy of the CT chromosome.

Relatedly, in the Fig. 2d legend, you refer to “cnv states per cell as a proxy”. Did you mean CNV states on chr2, where the CT event is? And do you mean to say as a proxy “for defining CT at this locus”?

Yes, thank you for pointing this out, this is exactly what we mean. We have revised the legend and caption of Figure 2.

Lines 306-320: The source of these fresh-frozen and FFPE samples should be included, as well as how CT status was determined in the fresh-frozen samples. In the methods (lines 809-10) it is stated that for the FFPE samples “TP53 mutation status was used as an approximation of chromothripsis status”. This should be reworded to state that TP53 mutation was taken to mean CT+, based upon a 100% CT rate in SHH TP53 mut medulloblastoma.

We have now added the source of the fresh-frozen (PMID: 28726821) and FFPE (PMID: 32296180 and PMID: 24651015) samples (see Methods, “Bulk RNA-seq Gene expression analysis”). The CT status in the fresh-frozen samples was determined based on the whole-genome sequencing data (Shatterseek with manual review, see Methods).

We have also reworded to state that “TP53 mutation was taken to mean CT+, based upon a close to 100% CT rate in SHH TP53 mut medulloblastoma”.

Supplementary Table 3.3: What are the two tabs? Are they meant to be the fresh-frozen and FFPE analyses? If so, this should be stated.

We have now clarified the content of the tables in the caption (see highlighted text).

Also, I noticed that SHH is missing from the data despite its mention (line 313), and that RRP1B has a $0 < \log_2FC < 1$, which should not be included based upon the criteria listed in lines 310-11.

We apologise for this oversight. The revised text as included in the rebuttal letter was accurate, however the main text was not revised accordingly. We have corrected the main text (page 12, line 312).

It should also be explicitly mentioned in the text that "it cannot be excluded that these differentially expressed genes are not due to TP53 status and not chromothripsis"^t, assuming TP53 status is how you determined CT status.

For the 46 FF samples, CT status was determined from whole-genome-sequencing (Shatterseek and manual review). Therefore, we are confident in the accuracy of the CT status as used for the differential expression analysis in the FF samples. Regarding the possibility to disambiguate the effect of *TP53* mutation and CT, the reviewer is correct that these two covariates are tightly coupled and hence we cannot consider them separately. We have revised the main text (page 25, paragraph 4), where we point to this.

Although compelling, the link between SETD2 and CT does not determine whether SETD2 causes chromothripsis or, whether like TP53mut, it allows CT cells to survive. To clarify this, lines 458-60 should be adjusted to say "...suggest a possible causative or permissive role for these two genes in the occurrence of CT in medulloblastoma." To further emphasize this, I would also change "linked" in line 526 with "correlated".

We have adopted the more careful wording suggested by the reviewer.

It is worth citing Espejo Valle-Inclan et al., Biorxiv 2023, as they have also shown detailed reports of subclonal chromothripsis in patient data, albeit not from single-cell data.

We have now added a reference to this study.

Reviewer #5 (Remarks to the Author):

Overall, I think the authors have done a reasonably good job addressing the concerns and criticisms raised by the previous reviewers. Importantly, the authors have used an orthogonal scDNA-seq method (Picoplex) and conducted extensive benchmarks and experiments (e.g., FISH, time-course LFS cell cultures) to verify the robustness of their methods and conclusions. Regarding the limited sample size, considering the rarity of LFS medulloblastoma, a significant expansion of the number of patients is impractical, and the current datasets seem sufficient for drawing the current conclusions in this paper. I have two suggestions to further improve the manuscript.

1. Regarding the title, "single-cell multi-omics" is misleading as the scDNA-seq and scRNA-seq were performed for different cells (although for the same sample). Single-cell multi-omics usually refers to simultaneous omic profiling for the same cell. This study actually also included plenty of bulk sequencing datasets, so I suggest to remove "single cell" from the title.

We thank the reviewer for their suggestion, and agree that “single-cell multi-omics” could indeed raise false expectations. Having said this, we feel that the single-cell aspect per se is a critical component of our study. We suggest the title

“Multi-omic and single-cell profiling of chromothriptic medulloblastoma reveals genomic and transcriptomic consequences of genome instability“

2. Regarding the authors' response to Main Concern #10 from Reviewer #2 (integration of bulk RNA-seq and scRNA-seq data), I think the impact of microenvironments on the differential expressions is not fully resolved. I suggest the authors to use Scissor (Sun et al. Nat Biotechnol 2021) which combines bulk RNA-seq to identify the phenotype-associated cell types in scRNA-seq data.

We understand the reviewer’s concerns and have tested the application of Scissor to our data. Before presenting these results, we would like to clarify our rebuttal to the previous reviewer. The basis of Main Concern #10 from Reviewer #2 was the application of the bulk signature to score single cells on a CT expression score. In particular, their concern was that the

“...approach cannot be used to identify chromothripsis-specific signatures that vary across single tumor cells.“

In response to this concern, we removed markers of non-malignant cell types from the DE, adjusted for tumour cell content, and presented the new results in the previous rebuttal. However, we did not emphasise that the scoring of single cells with our bulk CT signature was no longer presented in the results of our revised manuscript, as we could not definitively rule out the influence of microenvironmental factors on the signature. Instead, we now only make observational claims that some of the dysregulated pathways were common between the bulk signature and the pathways dysregulated between our genetic clones. Unfortunately, we had not removed the corresponding methods paragraph from the revised manuscript, which has now been removed from the current version. We apologise for this misunderstanding and thank the reviewer for pointing this out.

Following the current suggestion, we investigated whether the Scissor method could validate that the differentially expressed genes reported in our manuscript are actually differentially expressed in tumour cells. In brief, we ran the Scissor algorithm using the Fresh Frozen bulk RNAseq cohort (where CT was assessed directly from the matched WGS), together with the scRNAseq of SHH medulloblastoma patients from Riemondy et al., to identify the single cells which most closely represent CT+ and CT- patient samples within this dataset. We followed the preprocessing and hyperparameter tuning suggestions of the authors of Scissor as per the vignette. Scissor identified 1036 CT-like and 518 Non-CT-like cells in this scRNA-seq dataset. Importantly, Scissor classified overwhelmingly malignant cells as CT-like, compared to a majority of non-malignant cells as representing the Non-CT phenotype (Rebuttal Table 1). We note that this is not necessarily indicating that the Non-CT samples are composed of non-malignant cells, but that relatively, expression patterns in CT samples resemble normal cells less than Non-CT samples.

	Non-CT-like	CT-like
Non-Malignant	69%	1%
Malignant	31%	99%

Rebuttal Table 1: Proportions of malignant and Non-Malignant cells selected by Scissor as representative of CT-like and Non-CT-like cells.

We then set out to validate whether the 18 differentially expressed genes observed in our bulk analysis were also differentially expressed within a purely malignant cell population; between CT-like and Non-CT-like malignant Scissor cells. Of the 18 differentially expressed genes we report in our manuscript, 11 were detected in the scRNAseq, 3 up-regulated genes and 8 down-regulated genes. All 3 upregulated genes (*GLI2*, *CLASP1* and *TSN*) from our bulk analysis were also expressed significantly higher in CT-like Scissor cells (Wilcoxon test, adjusted p-value < 0.05). 5/8 of the bulk-downregulated genes were also expressed significantly lower in CT-like Scissor cells (Wilcoxon test, adjusted p-value < 0.05). In total, 8/11 of the reported bulk differentially expressed genes were also differentially expressed between the scissor selected populations (Rebuttal Figure 1). We then used the union of FF and FFPE results to score each Scissor cells for expression of the CT upregulated and downregulated genes, and confirmed that the up signature was expressed higher in Scissor CT-like cells and the down signatures was expressed higher in Scissor Non-CT-like cells (Rebuttal Figure 2).

Rebuttal Figure 1: Expression (Seurat normalized) for bulk-identified CT markers in Scissor selected Non-CT-like and CT-like malignant cells. * indicates that the differences are significant according to the Mann-Whitney U test, NS = Not Significant.

Rebuttal Figure 2: Gene Module scores as calculated by Seurat for the CT-up and CT-down genes identified by bulk RNAseq, in Scissor identified CT-like and Non-CT-like malignant cells.

Unfortunately, the main limitation of this analysis comes from the uncertainty that the CT-like cells selected in the scRNAseq from Riemondy et al. are faithful representations of a CT phenotype. Riemondy et al. sequenced only 9 SHH-subtype samples, and we cannot guarantee that any of these tumours were actually chromothriptic. Given the strong concerns by other reviewers about the specificity of the differential expression analysis to chromothripsis, we would prefer to refrain from presenting this analysis in the results section and limit our claims to the informal comparison currently present in the results.

Reviewer #6 (Remarks to the Author):

I carefully read your paper and rebuttal letter and I wish to acknowledge that you have addressed key concerns raised by the reviewers about this research in the revised version of your manuscript.

We thank the reviewer for their positive feedback.

REVIEWERS' COMMENTS

Reviewer #1 (Remarks to the Author):

I have reviewed the manuscript and the authors have appropriately addressed my concerns. One comment that does not require re-review: In Fig. 2c and Supplementary Fig. 3.6, percent should be written out of 100, not out of 1.

We thank the reviewer for pointing out these oversights and their comments throughout the process.

Reviewer #5 (Remarks to the Author):

My concerns have been well addressed.

We thank the reviewer for their comments throughout the process.